# COMiX: Compositional Explanations using Prototypes

## Abstract

Aligning machine representations with human understanding is key to improving interpretability of machine learning (ML) models. When classifying a new image, humans often explain their decisions by decomposing the image into concepts and pointing to corresponding regions in familiar images. Current ML explanation techniques typically either trace decision-making processes to reference prototypes, generate attribution maps highlighting feature importance, or incorporate intermediate bottlenecks designed to align with human-interpretable concepts. The proposed method, named COMiX, classifies an image by decomposing it into regions based on learned concepts and tracing each region to corresponding ones in images from the training dataset, assuring that explanations fully represent the actual decision-making process. We dissect the test image into selected internal representations of a neural network to derive prototypical parts (primitives) and match them with the corresponding primitives derived from the training data. In a series of qualitative and quantitative experiments, we theoretically prove and demonstrate that our method, in contrast to *post hoc* analysis, provides fidelity of explanations and shows that the efficiency is competitive with other inherently interpretable architectures. Notably, it shows substantial improvements in fidelity and sparsity metrics, including $48.82\%$ improvement in the C-insertion score on the ImageNet dataset over the best state-of-the-art baseline.

## 1 Introduction

Neural networks (NNs) have been successfully applied across various computer vision tasks, achieving notable results in safety-critical domains such as medical image classification (Huang et al., 2023), autonomous driving (Geiger et al., 2012), and robotics (Robinson et al., 2023) amongst others. However, explaining their decisions remains an ongoing research challenge (Samek et al., 2021).

The two key factors in interpreting neural network decisions are: (1) representing the reasoning behind the prediction in human-understandable terms and (2) ensuring that the explanations accurately reflect the underlying computations of the neural network. Beyond their face value, such interpretations can also help meet the legal requirements. The recently adopted EU AI Act (EUA, 2024) mandates that individuals should fully understand high-risk AI systems, enabling them to monitor these systems effectively, specifically requiring the ability to *'correctly interpret the high-risk AI system's output'*.

Most existing explanation methods address this problem using attribution-based techniques, which highlight the parts of the input that contribute to a particular decision (Selvaraju et al., 2017; Chattopadhay et al., 2018; Omeiza et al., 2019). However, these methods lack reliability as their explanations have been shown to be sensitive to factors which do not contribute to the model prediction (Kindermans et al., 2019). To address this issue, concept- and prototype-based explanations have been proposed, which aim to link the decision to examples that illustrate the underlying concepts (Kim et al. (2018); Ghorbani et al. (2019); Koh et al. (2020); Tan et al. (2024)). Nevertheless, such explanations have also been demonstrated to be insufficient for human understanding as they do not point to the reasons why the input is linked to the associated concept prototypes (Kim et al., 2016).

Studies of human understanding show that concepts can be decomposed into smaller constituents representing particular properties. These subconcepts can then be exemplified by the individual instances called *prototypes* (Murphy, 2004). In this work, we propose a concept-based interpretable-by-design method, which highlights common class-defining features between the input image and the

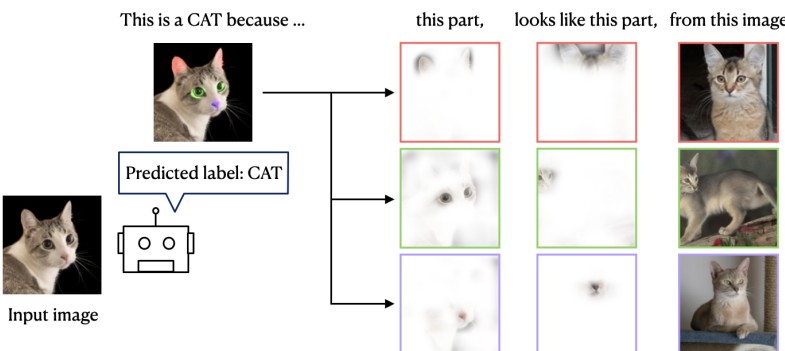

Figure 1: Humans often make sense of new or complex objects by comparing their parts to previously encountered prototypes (Smith et al. (1974)). For example, when describing something unfamiliar, people tend to point out resemblances between parts of the new object and familiar prototypes by stating that 'this part of the object looks like that other one I have seen before'. We propose a method to classify an image by decomposing it into regions based on learned concepts and tracing each region to the corresponding regions in images from training datasets. We refer to such interpretations as to *'COMiX panels'*

samples in the training dataset. This approach goes beyond attribution map predictions and presents a model, by design, that traces the decision to the original training data. Such decision-making process can be motivated by a number of safety-critical applications, for example, medical data analysis, where a doctor wants to find out the aspects that make this image similar to the previous ones.

We illustrate the idea of the proposed method, called COMiX, in Figure 1. For every test sample, we predict the output by linking them to a set of features in the training data. This link, by design, provides interpretations through the relationship between the testing image and the samples from the training set. This idea also extends to counterfactual interpretations, which demonstrate how the test sample relates to the classes that the model did not predict. It can also address the diagnostics of the misclassification cases, attributing the misclassification to the training data conditioned on class-defining features. We follow the convention from Rudin (2019) which contrasts *post hoc* explainability with *ante hoc* interpretability. COMiX is not *post hoc* and the interpretability comes from the decision-making. We formulate the following desiderata and demonstrate, in sections 3.3 and 4.2 how COMiX meets the demands of:

- **Fidelity**: The method should faithfully and wholly reflect the decision-making procedure, which is achieved by-design.

- **Sparsity**: For meaningful interpretation, the given class should activate only a handful of concepts. We enforce sparsity by restricting the decision-making to class-defining features. We also measure sparsity in Section 4.2 against the standard ViT (Dosovitskiy et al., 2021) baseline.

- **Necessity**: The concept is important for making the decision and its presence in the input is necessary. We evaluate this using the causal matrices (Petsiuk et al., 2018) in Section 4.2.

- **Sufficiency**: The concept presence in the input is sufficient for making the given decision. We present the proof in Section 3.3

The contributions of our paper are as follows:

- We propose a novel method, called COMiX, which reliably points prototypical regions in a testing image and matches them to regions in training images.

- Based on this method, we demonstrate how this method can be built upon existing inherently interpretable architectures with an additional value of concept discovery.

- We demonstrate, in a number of settings, the efficiency of COMiX through a series of qualitative and quantitative experiments, showing the advantages of the method over existing baselines in terms of fidelity and sparsity.

## 2 RELATED WORK

**Explainable and interpretable AI.** The early methods for neural network *post hoc* explanations, such as the work by Simonyan et al. (2013) and Grad-CAM (Selvaraju et al. (2017)), were grounded in the idea of differentiating through the model. Other important backpropagation-based models include Bach et al. (2015); Sundararajan et al. (2017). Perturbation-based methods, such as Ribeiro et al. (2016); Lundberg & Lee (2017); Petsiuk et al. (2018); Štrumbelj & Kononenko (2014), use perturbations to figure out input features' contributions. However, such a line of research is limited in its ability to capture the true inner workings of the original model (Rudin, 2019). To address this concern, a number of by-design interpretable machine learning models have been proposed, presenting the interpretable architectures (Böhle et al. (2022; 2024)), concept-bottleneck models (Koh et al. (2020); Shin et al. (2023); Schrodi et al. (2024); Losch et al. (2019); Qian et al. (2022)) and prototype-based interpretations (Chen et al., 2019; Donnelly et al., 2022; Angelov & Soares, 2020). Fel et al. (2023b) tackles a similar problem to the one in this paper: first, automatic extraction of concepts and then highlighting the similarities between such concepts and the testing image. However, the main conceptual difference between Fel et al. (2023b) and COMiX is that this work aims for by-design explanation of the decision-making while Fel et al. (2023b) addresses the problem of *post hoc* analysis. In contrast to these works, the described method is both inherently interpretable and offers interpretation through the training data.

**Concept discovery.** Closely related to the studied problem interpretation is the challenge of concept discovery, motivated by the neuroscience studies in human reasoning (Bruner et al., 1957). Kim et al. (2018) proposed a paradigm of concept activation vectors. Another study by Ghorbani et al. (2019) proposes extracting visual concepts through segmentation. Concept bottleneck models (Koh et al., 2020; Shang et al., 2024; Sheth & Ebrahimi Kahou, 2023; Havasi et al., 2022) introduce constraints into training so that the classifier is limited to using human-understandable features. Similar to these models, COMiX also leverages concept discovery, where the concepts are individual interpretable classifier features. On the contrary, we do not constrain the classifier to learn the human-understandable features and instead project the learned features into human-understandable space. In addition, COMiX traces these concepts back to the training data and provides inherent, by-design, interpretations, which have not, to the best of our knowledge, provided in the existing literature. ProtoPNet method (Chen et al. (2019)) is a well-known baseline for concept discovery through patch prototypes. It has been further developed in a number of works such as Donnelly et al. (2022); Ma et al. (2024); Sacha et al. (2023); Hase et al. (2019). Tan et al. (2024) propose to combine *post hoc* explainability methods with transparent concept-based reasoning. Bontempelli et al. (2022) analyses the problem of attainment of confounders within ProtoPNet and addresses it with human-in-the-loop model debugging.

**Evaluation of interpretability.** Hesse et al. (2023) propose a synthetic dataset and benchmark for part-level analysis of explainable models for image classification. Fel et al. (2023a) propose a set of metrics for explainable AI which assesses the quality of attribution-based explanations. They use the Insertion and Deletion metrics from Petsiuk et al. (2018) for attribution assessment. Important desiderata for concept extraction include sparsity of the outputs: not only do these outputs need to faithfully reflect the decision-making, but only a handful of concepts need to be activated for every testing image. To measure this ability, we leverage the metrics from the sparsity literature. Diao et al. (2022) propose a new PQ index metric, which measures the representation sparsity. One of the aspects, however, is that most of these metrics target the problem of attribution-based explanations. In our case, however, we combine concept-based and inherent attribution-based explanations, which allows us to evaluate the results using both C-insertion and C-deletion as well as the sparsity of concepts.

## 3 COMPOSITIONAL EXPLANATIONS USING COMiX

An overview of COMiX is presented in Figure 2. The figure demonstrates an example where a single Class Defining Feature (CDF) is used for prediction. For every test image, the final decision-making step aligns with human-interpretable reasoning: *'This image is classified as a dog because this region of the image resembles the corresponding region of this training image'*. This explanation fully corresponds to the underlying computations, providing a faithful and complete representation

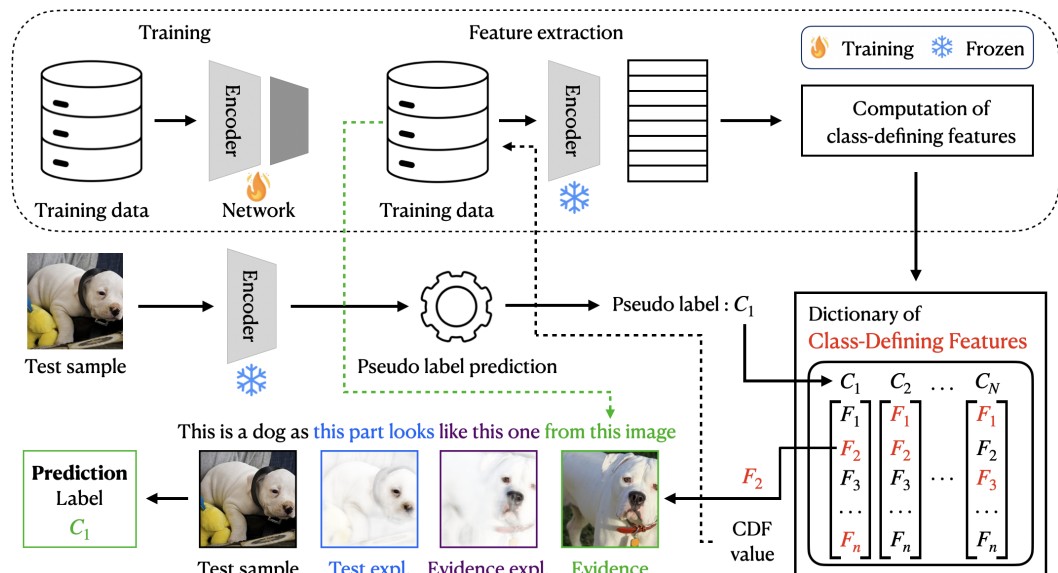

Figure 2: COMiX method overview.

of the decision process, i.e. not an approximation of the computation. We train a B-cos network, an inherently interpretable model, on the training data. Using the train features from this encoder, we compute the CDFs. During inference, we project the test image into the CDF space using a pseudo-label. For each CDF feature, we retrieve the closest matching training data point. Projecting the CDF features into image space allows us to localize the prototypical regions in the test image that correspond to the training data. The final prediction is obtained through majority voting of the labels assigned by each CDF feature.

### 3.1 PRELIMINARIES: B-COS ARCHITECTURE

A B-cos encoder generates a reliable explanation of its computation. B-cos networks are neural networks in which all the linear layers (along with activations) are replaced by B-cos layers. For more details on the formulation and training of these networks, we refer the reader to Böhle et al. (2022; 2024) and to Appendix A. Operation of a B-cos layer at a node for an input $\mathbf{x}$ and weights $\mathbf{w}$ leading to the node is given by

$$\text{B-cos}(\mathbf{x}; \mathbf{w}) = \|\mathbf{x}\| \cdot \|\mathbf{w}\| \cdot |\cos(\angle(\mathbf{x}, \mathbf{w}))|^B \cdot \text{sign}\left(\cos(\angle(\mathbf{x}, \mathbf{w}))\right), \quad (1)$$

where $B$ is a hyper-parameter that influences the extent to which alignment between $\mathbf{x}$ and $\mathbf{w}$ contributes to the magnitude of the output. Replacing linear layers with B-cos layers removes the need for other explicit non-linearity while training the network. Given an input, B-cos layer becomes a linear layer followed by a scalar multiplication (the cosine score: Equation 1). As each layer becomes a linear operation, the neural network collapses into a single linear transform that faithfully summarises the entire model computations. Moreover, the B-cos layers introduce alignment pressure on their weights during optimization. For the output of a node to be high, the input must align well with the node's incoming parameters, indicated by a high value of $\cos(\angle(\mathbf{x}, \mathbf{w}))$. In short, we choose the B-cos network for two reasons: (a) B-cos has an input-dependent non-linearity which collapses the encoder computations into a linear transformation for a test sample, and (b) the collapsed linear operation (i.e., matrix) is aligned to the input sample when the output is high.

Given an input image $\mathbf{x}$, $(L + 1)$-layer B-cos network collapses into a linear layer. This matrix is aligned with the input if the output is high. The $(L + 1)$-layer transformation can be presented as a shortcut representation

$$W_{1 \rightarrow (L+1)}(\mathbf{x}; \theta) = W_{(L+1)} \circ W_L ... \circ W_1(\mathbf{x}; \theta), \quad (2)$$

The final output is obtained as

$$f(\mathbf{x}; \theta) = W_{1 \rightarrow (L+1)}(\mathbf{x}; \theta)\mathbf{x}, \quad (3)$$

We modify the previous formulation of B-cos to get the explanation of features that are activated by the input. Previous work has also shown that B-cos transformers inherently learn human-interpretable features. We compute the explanation for a feature $i$ in the $L^{\text{th}}$ layer as $W_{1 \to L}(\mathbf{x}; \theta)^i$.

## 3.2 Compositional Explanations using Prototypes (COMiX)

We present the complete methodology in Algorithm 1. Hereafter $\arg_x \text{top}_k[\cdot]$ denotes the generalisation of the $\arg\max_x[\cdot]$ operator where the maximum is replaced with top $k$ values. The algorithm starts (**Step 1**) with calculation of embeddings $W_{1 \to L}(\mathbf{x}; \theta)$ (Encoder stage in Figure 2). It proceeds with the pseudo-label prediction (**Step 2**), and the selection of the CDFs for a given pseudo-label (**Step 3**). The per-feature predictions are calculated from the CDFs for top $M$ CDFs for the pseudo-label and $K$ nearest neighbours (**Step 4**). In **Step 5**, we calculate the corresponding explanations. It is important to see that instead of one label the method gives a number of predictions, one per every feature and per every nearest neighbour. Further in the experimental section, we calculate the aggregated prediction as a mode of the prediction set $G(\mathbf{x}; \theta)$.

---

**Algorithm 1:** Compositionally explainable classifier COMiX

---

**Data:** Image $\mathbf{x}$; training dataset $\mathcal{D}$;

    class-defining features $P^C = \{P^c \; \forall c \in \mathbb{C}\}$; a number of features $M$ to be explained;

    a number of nearest neighbours $K$

**Result:** $M \times K$ per-feature predictions $G(\mathbf{x}, \theta) = \{g^i, j(\mathbf{x}; \theta)\}_{i \in [1 \ldots P^C], j \in [1, K]}$;

    explanations $E(\mathbf{x}; \mathcal{D}, P^C)$ for retrieved concepts

1. Calculate $W_{1 \to L}(\mathbf{x}; \theta)$ as per Equation (2)

2. Predict the nearest-neighbour pseudo-label class using Equation (5)

3. Using the pseudo-label, select the top $M$ scalar class-defining features $P^{\tilde{g}(\mathbf{x}; \theta)}$ (see Equation (7))

4. Calculate the per-feature predictions $G(\mathbf{x}; \theta, P^C)$ from the class-defining features $P^{\tilde{g}(\mathbf{x}; \theta)}$ according to Equations (9) and (10)

5. Calculate the explanations for the $K$ nearest neighbours for every class-defining feature according to Equation 11

---

We define a *training dataset* $\mathcal{D} = \{\mathbf{d}_1, \mathbf{d}_2, \ldots \mathbf{d}_n\}$ which contains a set of reference image samples, annotated by the labels $\mathcal{L} = \{\mathbf{l}_1, \mathbf{l}_2, \ldots \mathbf{l}_n\}$ from a label set $\mathbb{C}$ as $l(\mathbf{d}_i) = \mathbf{l}_i \; \forall i \in [1, n]$.

We focus our experiments on the final layer and analyse its properties through the lens of transformation $W_{1 \to L}$, which has shape $C_L \times (W \cdot H \cdot D)$. Here $C_L$ is the number of features in the last layer ($L^{\text{th}}$ layer). The per-feature attribution explanations for a given input $\mathbf{x}$ is given by $\mathbf{s}_L^\cdot(\mathbf{x}; \theta)$ defined as follows:

$$W_{1 \to L}(\mathbf{x}; \theta) = \left( \mathbf{s}_L^1(\mathbf{x}; \theta), \mathbf{s}_L^2(\mathbf{x}; \theta), \ldots, \mathbf{s}_L^{C_L}(\mathbf{x}; \theta) \right)^T, \tag{4}$$

**Step 2** uses the following equation to compute the pseudo-label class:

$$\tilde{g}(\mathbf{x}; \theta) = l(\arg\min_{\mathbf{d}} \{ \ell^2(W_{1 \to L}(\mathbf{x}; \theta)\mathbf{x}, W_{1 \to L}(\mathbf{d}; \theta)\mathbf{d}) \; \forall \mathbf{d} \in \mathcal{D}\}) \tag{5}$$

**In Step 3,** for the dataset $\mathcal{D}$, we calculate the top $M$ scalar *class-defining features* $P^c$ for class $c \in \mathbb{C}$ by using maximum mutual information:

$$F = \{W_{1 \to L}(\mathbf{d}, \theta)\mathbf{d} \; \forall \mathbf{d} \in \mathcal{D}\}, F_j = \{\mathbf{s}_L^j(\mathbf{d}, \theta)\mathbf{d} \; \forall \mathbf{d} \in \mathcal{D}\}, \tag{6}$$

$$P^c = \{\arg_j \text{top}_M I(F_j, l(F_j) = c) \; \forall j \in [1 \ldots C_L]\}, \tag{7}$$

Figure 3: Examples of *COMiX panel* interpretations for Oxford-IIIT Pets (left) and CUB-200-211 dataset (right).

where $c \in \mathbb{C}$ is a label for class $c$, $l(F_j)$ is a ground-truth label operator for the feature $F_j$, and the mutual information $I(X, Y)$ is defined as

$$I(X, Y) = \sum_{\langle x, y \rangle \in \langle X, Y \rangle} p(x, y) \log \left( \frac{p(x, y)}{p(x)p(y)} \right). \tag{8}$$

The introduction of pseudo-labels is necessary for the selection of a small number of CDFs and therefore restricting the explanation to a small number of features. They constitute the initialisation for the decision-making process, which allows bootstrapping the selection of class-defining features.

**Step 4** calculates the per-feature predictions $G(\mathbf{x}; \theta, P^C)$ through the following equations:

$$G(\mathbf{x}; \theta, P^C) = l(\mathbf{D}^*(\mathbf{x}, \theta, P^C)), \tag{9}$$

$$\mathbf{D}^*(\mathbf{x}, \theta, P^C) = \{\arg_{\mathbf{d}} \text{top}_K \{-\ell^2([W_{1 \to L}(\mathbf{x}; \theta)\mathbf{x}]_f, [W_{1 \to L}(\mathbf{d}; \theta)\mathbf{d}]_f) \, \forall \mathbf{d} \in \mathcal{D}\}\}_{f \in P^{\tilde{g}(\mathbf{x};\theta)}} \tag{10}$$

**In Step 5,** explanations for the CDF are calculated using the following equations:

$$E(\mathbf{x}; \mathcal{D}, P^C) = E(\mathbf{x}; \mathcal{D}, P^{\tilde{g}(\mathbf{x};\theta)}) = \{\langle s_L^i(\mathbf{x}_i, \theta), s_L^i(\mathbf{d}_i^{\text{nearest}}, \theta) \rangle \, \forall i \in P^{\tilde{g}(\mathbf{x};\theta)}\}, \tag{11}$$

where the training samples' features, nearest to a class-defining feature of the testing image $\mathbf{x}$, are calculated as $\mathbf{d}_i^{\text{nearest}} = \arg_{\mathbf{d}} \text{top}_K \{\ell^2((W_{1 \to L}(\mathbf{d}; \theta)\mathbf{d})_i, (W_{1 \to L}(\mathbf{x}; \theta)\mathbf{x})_i), \forall \mathbf{d} \in \mathbf{D}^*\}$ and $s_L^i(\mathbf{d}, \theta)$ is $i$-th row of $W_{1 \to L}(\mathbf{d}, \theta)\}$.

## 3.3 DEMONSTRATION OF MEETING THE DESIDERATA

We define the criterion of sufficiency of the explanation and demonstrate how and in which conditions we meet this criterion. In Table 5 the experimental section, we also outline how COMiX addresses the requirements of **sparsity**. We address the question of **fidelity** experimentally, by measuring insertion and deletion metrics in Section 4.2.

We address **necessity** (i.e., presence of the elements of the explanation necessary for the decision making) of the explanations $E(\mathbf{x}, \mathcal{D}, P^C)$ from Equation 11 by visualising the elements of exact same nearest-neighbour samples that are present in the decision-making procedure in Equation 10.

We define **sufficiency** of the explanations $E(\mathbf{x}; \mathcal{D}, P^C)$ in a way that the same explanation would imply the same output:

$$\forall \mathbf{x}, \mathbf{x}' E(\mathbf{x}'; \mathcal{D}, P^C) = E(\mathbf{x}; \mathcal{D}, P^C) \implies G(\mathbf{x}'; \theta, \mathcal{D}, P^C) = G(\mathbf{x}; \theta, \mathcal{D}, P^C) \tag{12}$$

**Theorem 1.** *Assume $\tilde{g}(\mathbf{x}; \theta) = g(\mathbf{x}; \theta) \, \forall g(\mathbf{x}; \theta) \in G(\mathbf{x}; \theta)$. Then the explanation $E(\mathbf{x}; \mathcal{D})$ is sufficient for the prediction $G(\mathbf{x}; \theta, \mathcal{D})$ according to Algorithm 1.*

*Proof.* Suppose that $E(\mathbf{x}'; \mathcal{D}, P^C) = E(\mathbf{x}; \mathcal{D}, P^C)$ and $G(\mathbf{x}'; \theta, \mathcal{D}, P^C) \neq G(\mathbf{x}; \theta, \mathcal{D}, P^C)$ for some $\mathbf{x}, \mathbf{x}'$. Using the assumption that $\tilde{g}(\mathbf{x}; \theta) = g(\mathbf{x}; \theta) \, \forall g(\mathbf{x}; \theta) \in G(\mathbf{x}; \theta)$, one can note that the two sets $\mathbf{D}^*(\mathbf{x}, \theta, P^C), \mathbf{D}^*(\mathbf{x}', \theta, P^C)$ cannot possibly be the same as the labels of the two sets are different and the same training datum $\mathbf{d}$ cannot have two different labels, i.e. $G(\mathbf{x}'; \theta, \mathcal{D}, P^C) \neq G(\mathbf{x}; \theta, \mathcal{D}, P^C)$ means that $\mathbf{D}^*(\mathbf{x}, \theta, P^C) \neq \mathbf{D}^*(\mathbf{x}', \theta, P^C)$. This means that the explanations $E(\mathbf{x}, \mathcal{D}, P^C)$ and $E(\mathbf{x}', \mathcal{D}, P^C)$ are calculated in Equation 11 over two different subsets of training samples and therefore cannot possibly be the same. Therefore, we can see that, by contradiction, Equation 12 holds true for Algorithm 1.

□

Table 1: Evaluation of performance against ProtoPNet (Chen et al. (2019)), B-Cos (Böhle et al. (2024)) and common deep-learning baselines on CUB-200-2011 (full images), the values denoted by $*$ are obtained from Donnelly et al. (2022)

| Architecture | Baseline | ProtoPNet | B-cos | COMiX |
|---|---|---|---|---|
| ResNet34 (He et al. (2016)) | 76.0∗ | 72.4* | 74.3 | 73.8 |
| ResNet152 (He et al. (2016)) | 79.2* | 74.3* | 76.5 | 76.2 |
| DenseNet121 (Huang et al. (2017)) | 78.2∗ | 74.0* | 73.6 | 73.2 |
| DenseNet161 (Huang et al. (2017)) | 80.0* | 75.4* | 76.1 | 76.1 |

Table 2: Evaluation of performance against B-Cos (Böhle et al. (2024)) and baseline ViT (Dosovitskiy et al. (2021)), $K$-NN refers to the baseline of B-cos + $K = 3$ nearest neighbours, pretrained on ImageNet

| Dataset | ViT | B-cos | $k$-NN | COMiX |
|---|---|---|---|---|
| Oxford-IIIT Pets | $90.32 \pm 0.03$ | $89.32 \pm 0.13$ | $89.23 \pm 0.11$ | $87.73 \pm 0.21$ |
| CUB-200-2011 | $79.62 \pm 0.04$ | $79.23 \pm 0.08$ | $78.98 \pm 0.06$ | $74.14 \pm 0.18$ |
| Stanford Cars | $90.72 \pm 0.32$ | $86.53 \pm 0.31$ | $87.95 \pm 0.24$ | $86.81 \pm 0.24$ |
| CIFAR-10 | $93.34 \pm 0.08$ | $93.10 \pm 0.15$ | $93.28 \pm 0.09$ | $91.21 \pm 0.19$ |
| CIFAR-100 | $78.61 \pm 0.03$ | $76.07 \pm 0.06$ | $74.23 \pm 0.04$ | $76.42 \pm 0.12$ |
| ImageNet | $78.90 \pm 0.24$ | $77.78 \pm 0.24$ | $75.16 \pm 0.24$ | $74.28 \pm 0.38$ |

## 4 EXPERIMENTS AND DISCUSSION

In this section, we evaluate COMiX through a series of quantitative and qualitative experiments. We assess the model's performance on standard benchmarks (accuracy, fidelity, and sparsity) to validate the method's effectiveness. We compare the accuracy of COMiX with other baseline methods. We also show the robustness of the model performance across different backbones. We demonstrate the fidelity of COMiX by evaluating the method using causal matrices (Ghorbani et al. (2019)). Additionally, we present qualitative analyses by visualizing the prototypical regions identified during inference, providing insights into the interpretability and decision-making process of the model. These experiments highlight the model's ability to provide transparent and faithful explanations while maintaining competitive accuracy.

Figure 3 shows the explanation generated by the method using only one evidence sample and one feature alone used for prediction. The second image in the panel shows the super-pixel like segmentation generated based on the dominant CDF feature for every pixel ($\arg\max_i \{\mathbf{s}_L^i(\mathbf{x}, \theta)\} \forall i \in (\mathbf{s}_L^1(\mathbf{x}, \theta), \cdots, \mathbf{s}_L^i(\mathbf{x}, \theta)), \cdots \mathbf{s}_L^{C_L}(\mathbf{x}, \theta))$. We present more interpretation examples in Appendix E.

### 4.1 DATASETS

**Datasets** We train and evaluate the presented model on a number of commonly-used computer vision datasets. CIFAR-10&100 (Krizhevsky et al., 2009) contain generic natural images from 10 and 100 diverse classes respectively. CUB-200-2011 (Welinder et al., 2010) is a commonly used dataset for evaluating interpretable vision models, which contains 200 fine-grained classes of birds. Stanford cars dataset (Krause et al., 2013) contains 196 classes of cars. Oxford-IIIT Pets (Parkhi et al., 2012) contains a fine-grained collection of images of 37 classes of cats and dogs. Finally, we present the results on ImageNet (ILSVRC 2012) (Russakovsky et al., 2015) which has 1000 diverse classes.

**Baselines** We compare COMiX to the following well-known baselines: (1) Standard architectures such as ResNet He et al. (2016), DenseNet Huang et al. (2017) and ViT Dosovitskiy et al. (2021) (2) the B-Cos counterparts of the aforementioned architectures (Böhle et al., 2022; 2024) and (3) a number of interpetable and explainable ML methods including ProtoPNet (Chen et al., 2019), Deformable ProtoPNet (Donnelly et al., 2022) and CAM (Zhou et al., 2016).

**Experimental setup** We pretrain the backbone B-cos models on ImageNet (Russakovsky et al., 2015) and then on target datasets. The details of the experimental setup, hardware configuration and hyperparameters are described in Appendix B.

Table 3: Interpretability vs accuracy, % (adopted from Donnelly et al. (2022), all values except from ViT, B-cos, and COMiX, come from there)

| Interpretability | Method | CUBS | Method | Cars |
|---|---|---|---|---|
| None | ViT (Dosovitskiy et al. (2021)) | 79.6 | ViT (Dosovitskiy et al. (2021)) | 92.72 |
| Part-level attention | TASN (Zheng et al. (2019)) | 87.0 | FCAN (Liu et al. (2016)) RA-CNN (Fu et al. (2017)) | 84.2 87.3 |
| Part-level attention + Prototypes | ProtoPNet (Chen et al. (2019)) Def. ProtoPNet (Donnelly et al. (2022)) | 81.1 86.4 | ProtoPNet (Chen et al. (2019)) Def. ProtoPNet (Donnelly et al. (2022)) | 77.3 86.5 |
| Object-level attention | CAM (Zhou et al. (2016)) CSG (Liang et al. (2020)) B-cos (Böhle et al. (2024)) | 63.0 78.5 79.2 | B-cos (Böhle et al. (2024)) | 86.5 |
| Object-level attention + Prototypes | COMiX | 74.0 | COMiX | 86.80 |

**Accuracy evaluation** Table 1 shows that COMiX provides competitive accuracy compared to the baselines of B-Cos/ViT on the full-frame CUB-200-2011 dataset (Welinder et al., 2010). In all cases, the accuracy is calculated as the mode of the values of output predictions $G(x, \theta)$. In this experiment, the B-cos architecture for both B-cos and COMiX mirrors the corresponding deep-learning baseline, i.e. B-Cos/ViT is a B-cos counterpart of the ViT model as described in Böhle et al. (2024). Table 1 does not list the ViT results. This is due to the reason that, as suggested in Xue et al. (2022), ProtoPNet cannot be used with the ViT architecture without substantial modifications.

In Table 2, we evaluate the performance of the model on a variety of datasets using just the B-cos/ViT backbone as outlined in Appendix B. Additionally, in Appendix C, we show that surprisingly, COMiX provides impressive capabilities for learning without finetuning on the target data, opening up the potential for adaptation of the method to the new datasets without finetuning. In this setting, the models were pretrained on ImageNet, and then, during the inference stage, matched by the nearest neighbours procedure as described in Algorithm 1. The $k$-NN results were calculated for $k = 3$ using the B-cos backbone.

In Table 3, we demonstrate the trade-off between interpretability and accuracy across the categories. Following Chen et al. (2019), we compare different types of explainable and interpretable models. Many of the *post hoc* attribution-based methods, as well as original B-cos model, provide interpretation through object-level attention, while patch-based methods such as, notably, ProtoPNet (Chen et al. (2019)) are referred to as part-level attention methods.

## 4.2 Evaluation of the Interpretability Properties

Table 4 presents the evaluation of Average Drop, Average Increase, C-insertion, and C-deletion metrics (Ghorbani et al. (2019)), which are the standard metrics to quantify fidelity of the explanations both for *ante hoc* (by-design) and *post hoc* methods. The average drop (increase) metrics are calculated as the drop (increase) in performance of prediction when we drop (add) $50\%$ of the pixels with the lowest (highest) attribution. The C-insertion and C-deletion metrics represent the area under the curve for insertion (deletion) of pixels in the increasing (decreasing) order of pixel attribution value.

The property of sparsity is crucial for selecting meaningful class-defining features. Low sparsity would mean that more CDF need to be selected to meaningfully represent the class. To measure sparsity, we use the PQ-Index sparsity measure (Diao et al. (2022)). For the vector $\mathbf{w} \in \mathbb{R}^d$ it

Table 4: Evaluation of Average Drop, %, Average Increase, %, C-insertion and C-deletion metrics (results marked with $^*$ are taken from Zeng et al. (2023))

| Method | Drop ↓ | Increase ↑ | C-insertion ↑ | C-deletion ↓ |
|---|---|---|---|---|
| Grad-CAM (Selvaraju et al. (2017)) | 41.5* | 20.8* | 0.4626* | 0.1110* |
| Grad-CAM++ (Chattopadhay et al. (2018)) | 40.8* | 22.3* | 0.4484* | 0.1179* |
| SGCAM++ (Omeiza et al. (2019)) | 41.1* | 23.4* | 0.4504* | 0.1169* |
| Score-CAM (Wang et al. (2020)) | 35.6* | 29.5* | 0.4929* | 0.1099* |
| Group-CAM (Zhang et al. (2021)) | 35.7* | 29.7* | 0.4930* | 0.1108* |
| Abs-CAM (Zeng et al. (2023)) | 34.2* | 30.1* | 0.4949* | **0.1096***  |
| COMiX | 41.3 | **36.5** | **0.7365** | 0.1214 |

Table 5: Evaluation of sparsity using PQ-index (larger is better) between COMiX and ViT (Dosovitskiy et al. (2021))

| Method/ Dataset | Oxford-IIIT Pets | CUB-200-2011 | Stanford Cars | CIFAR-10 | CIFAR-100 |
|---|---|---|---|---|---|
| ViT | 0.21 | 0.42 | 0.61 | 0.63 | 0.45 |
| COMiX | **0.26** | **0.52** | **0.67** | **0.65** | **0.48** |

is defined as $I_{p,q}(\mathbf{w}) = 1 - d^{q^{-1} - p^{-1}} \frac{|\mathbf{w}|_p}{|\mathbf{w}|_q}$, where $|\mathbf{w}|_q > 0$ is a $\ell^q$ norm, $0 < p \leq 1 < q$ are hyperparameters. In our analysis, we use the values $p = 1, q = 2$.

In Table 5, we measure the backbone B-cos/ViT architecture sparsity for the extracted features $W_{1 \to L}(\mathbf{x}, \theta)\mathbf{x}$ against its standard ViT counterpart, trained on the target data. We found that COMiX, based on B-cos/ViT features, benefits from better feature sparsity compared to the ViT baseline, which justifies the use of class-defining features.

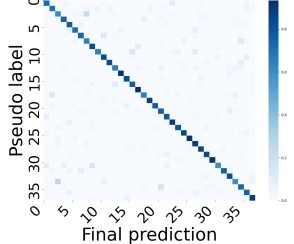

**Analysis of the impact of pseudo-labels**   In Figure 4, we present the confusion matrices for the Oxford-IIIT Pets dataset, which compares the final prediction vs pseudo-labels. The extended version of this figure which compares between the true labels, the final predictions, and the pseudo-labels is included in Appendix F.

Figure 4: Final prediction vs pseudo-label confusion matrix on Oxford-IIIT Pets dataset

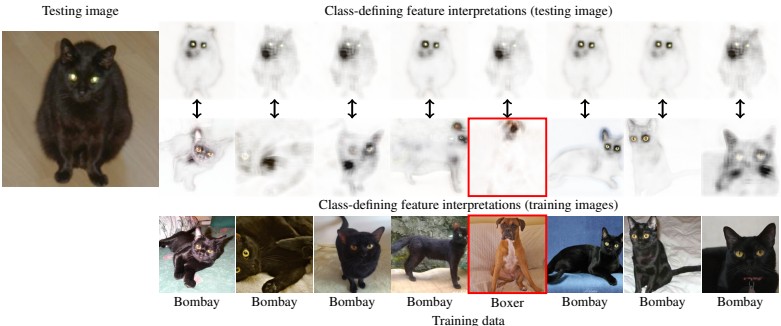

Figure 5: Interpretation for a sample image from the Oxford-IIIT Pets dataset: the model correctly classifies the input image as 'Bombay cat'. This visualization demonstrates the similarity between the test image and seven training images of the 'Bombay cat' class and one image of a boxer dog (highlighted in red), offering insight into the model's decision-making process.

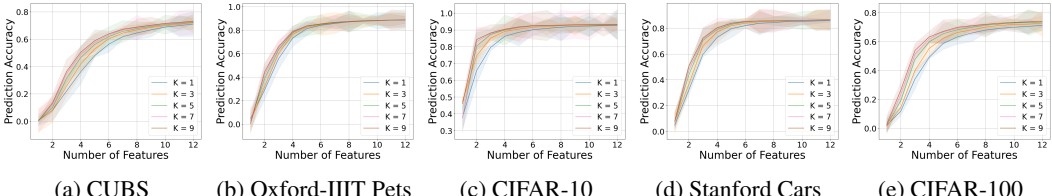

|     |     |     |     |     |
| --- | --- | --- | --- | --- |
| (a) CUBS | (b) Oxford-IIIT Pets | (c) CIFAR-10 | (d) Stanford Cars | (e) CIFAR-100 |

Figure 6: The performance of COMiX with different choice of hyperparameters. Each of the curves corresponds to a value of $K$ ($K$ nearest neighbours), and the horizontal axis shows the number of features $M$ used for prediction.

### 4.3 DEMONSTRATION OF PROTOTYPICAL EXPLANATION

We qualitatively show the examples for the prototypical explanation in Figure 5. In this figure as well as in Appendix E, we demonstrate that on a number of use cases, the model can present both factual and counterfactual interpretations for a number of complex scenarios. The reader can note the correspondence between the features of the training and testing image attribution maps arising from explanations by class-defining features.

### 4.4 PARAMETER SENSITIVITY ANALYSIS

In Figure 6 we compare the performance of the method depending upon the number $K$ of nearest neighbours as well as the number of features $M$ used for the prediction. This analysis shows that there is an inherent trade-off between the performance and the conciseness of explanation. In Appendix D we also show similar experimental results in a setting similar to the state-of-the-art work. In this setting, instead of performing the predictions feature-by-feature as in Algorithm 1, Step 4, we show the prediction performance using $\ell^2$ distances over the whole set of CDFs as common in the current literature. This creates another trade-off: while it may increase the performance, the downside of such an alternative approach would be that the explanations, and the decisions, would not follow directly from the given features but from their combination.

## 5 CONCLUSION

COMiX demonstrates a novel form of interpretable machine learning, which performs decision-making through the similarity of the concepts within the test image to the corresponding concepts in the training set. We demonstrate that this allows both factual (*Why did the model predict this class?*) and counterfactual analysis (*How would the model explain the predictions if the alternative class was predicted?*). The experimental results show both a competitive accuracy of the method and demonstrate, empirically and theoretically, that the method has favourable properties of fidelity, necessity, sufficiency, and sparsity. Surprisingly, it also demonstrates impressive finetuning-free $k$-NN generalisation to new datasets.

It is also worth noting that the training dataset $\mathcal{D}$ from Algorithm 1 can be, without any changes in the method, be replaced with a trusted dataset, which can contain a private collection of data which the predictor could relate the testing image to. It can be motivated by both lack of access to training images or by lack of trust in the training data due to an inherent noise and can be especially useful in safety-critical domains such as medical imagery.

While the current work only focuses on image classification, future work can expand this method to the segmentation scenarios. To reflect upon this, we demonstrate, in Appendix G, the potential for concept-based segmentation using COMiX. Such concept-based segmentation can enable human-in-the-loop decision-making: a human can change class-defining features in the model by selecting the corresponding segments. One can see the potential of the method to detect manipulated imaging and adversarial attacks by highlighting the areas of forgery and comparing them with training-set examples. We outline the limitations and broader impacts in Appendices H and I respectively.

## 6 Reproducibility Statement

To ensure reproducibility of the results presented in this paper, we provide detailed information on the following key aspects:

- **Datasets:** The experiments were conducted on well-known datasets, including CIFAR-10, CIFAR-100, CUB-200-2011, Stanford Cars, and Oxford-IIIT Pets. These datasets are publicly available and widely used in computer vision research.

- **Model Architectures:** We used B-cos (Böhle et al. (2022)) and ViT models, which are described in detail in both the main text and the Appendix. The model architecture, including any modifications for our method (COMiX), is fully explained, ensuring that the implementation can be reproduced by others. The training details are given in the Appendix 6.

- **Training Details:** The model training process is described with exact hyperparameters provided in Appendix B. We also offer details on hardware used (e.g., GeForce RTX 2080 Ti with CUDA version 12.5) and software packages (e.g., NumPy, PyTorch, Torchvision), making it easy to replicate the experiments.

- **Evaluation Metrics:** All metrics used for evaluation, such as accuracy, fidelity, sparsity, and C-insertion/C-deletion metrics, are well-documented, ensuring consistency in reproducing the reported performance.

- **Code Availability:** To support reproducibility, we will provide the code used to conduct these experiments, including tables and analysis. This code, including all prompt templates and post-processing scripts, will be made publicly available upon publication.

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

# A B-COS NETWORKS

B-cos networks offer a novel approach to improving the interpretability of deep learning models by ensuring that input features align with the model's weights throughout training. This innovation arises from the realization that although deep neural networks excel in performance across various tasks, their internal workings remain largely opaque and hard to interpret. Typically, deep models rely on linear transformations coupled with non-linear activations, a design that contributes to their "black box" nature.

In contrast, B-cos networks replace traditional linear transformations with the B-cos transformation, which promotes alignment between inputs and weights. This transformation is defined as

$$\text{B-cos}(\mathbf{x}; \mathbf{w}) = \|\mathbf{w}\|\|\mathbf{x}\| \cos^B(\theta) \cdot \text{sign}(\cos(\theta)), \tag{13}$$

where $\theta$ denotes the angle between the input vector $\mathbf{x}$ and the weight vector $\mathbf{w}$, and $B$ is a hyperparameter that amplifies the model's sensitivity to alignment. This transformation shifts the model's focus from merely achieving high performance to fostering interpretability by emphasizing the relationship between the input data and model features.

The training of B-cos networks integrates this alignment directly into the optimization process. By applying alignment pressure during weight adjustment, B-cos networks encourage the model to align its weights with the most relevant input features, making this alignment a key objective rather than a byproduct of training, which is a departure from conventional methods focused solely on minimizing prediction error.

The integration of B-cos transformations into existing architectures is seamless since they can serve as direct replacements for typical linear layers. This compatibility enables the application of B-cos networks to a broad array of architectures such as VGG, ResNet, InceptionNet, DenseNet, and Vision Transformers (ViTs) Böhle et al. (2024), without significant changes to their core structure. Empirical results demonstrate that this transformation maintains competitive performance on standard datasets like ImageNet while enhancing model interpretability.

During inference, a key advantage of B-cos networks becomes apparent. The sequence of B-cos transformations throughout the network simplifies into a single linear operation, as the successive alignment-focused layers collapse into a single transformation. Mathematically, this is expressed as:

$$W_{1 \to L}(\mathbf{x}) = W_1 \times W_2 \times \ldots \times W_L, \tag{14}$$

where $W_{1 \to L}(\mathbf{x})$ represents the effective weight matrix over all $L$ layers, and $W_1, W_2, \ldots, W_L$ are the weight matrices of the individual B-cos layers. This reduces the network's computation at test time to:

$$y = W_{1 \to L}(\mathbf{x}) \cdot \mathbf{x}, \tag{15}$$

where $y$ is the output. This reduction to a single matrix-vector multiplication significantly improves both computational efficiency and transparency, offering a clear view of how input features affect the output. The network's behavior, represented by $\theta_{1 \to L}$ in the main text, becomes fully interpretable, as emphasized in Böhle et al. (2022).

# B EXPERIMENTAL SETUP

## B.1 HARDWARE AND SOFTWARE CONFIGURATION

We trained and tested our models using GeForce RTX 2080 Ti with CUDA version 12.5. In our work we use the following software packages for Python:

1. NumPy 1.26.2

2. PyTorch 2.1.2

3. Torchvision 0.16.2

4. Matplotlib 3.8.2

Table 6: Hyperparameters of the B-cos model training

| Hyperparameter | Value | Comments |
|---|---|---|
| Learning rate | 0.01 | |
| Max # epochs | 500 | with early stopping |
| Batch size | 16 | |
| Drop out value | 0.5 | |
| B-value in B-cos | 1.5 | |
| Loss | BCE Loss | |
| Final activation | sigmoid | |

Table 7: Comparison between the performance of finetuned and non-finetuned model based on B-cos/Vit backbone

| Dataset | COMiX | COMiX (no finetuning) |
|---|---|---|
| Oxford-IIIT Pets | 87.73 | 85.52 |
| CUB-200-2011 | 74.14 | 70.04 |
| Stanford Cars | 86.81 | 84.03 |
| CIFAR-10 | 91.21 | 90.04 |
| CIFAR-100 | 76.42 | 72.92 |

## B.2 MODEL TRAINING AND EVALUATION DETAILS

Except for zero-shot learning settings, the B-cos models has been trained with the hyperparameters outlined in Table 6. For more details on the meaning of the training parameters for the B-cos model, please follow the work Böhle et al. (2024).

For the pretrained baselines and models, we use the models from the open sources, which can be downloaded from the following repository: https://github.com/B-cos/B-cos-v2. For B-cos/ViT, we use `vitc_l_patch1_14` model pretrained on ImageNet

## C NON-FINETUNED MODEL PERFORMANCE

Table 7 demonstrates comparative performance between the finetuned and non-finetuned method.

## D FURTHER DETAILS ON HYPERPARAMETER CHOICE

In Figure 8 we present the hyperparameter sensitivity analysis graphs for the $\ell^2$ distances between the whole set of CDFs. The experimental scheme is given in Figure 7.

## E ADDITIONAL QUALITATIVE RESULTS

In Figures 9, 10, 11, we show additional qualitative results.

## F CONFUSION MATRICES FOR PSEUDO-LABELS

In Figure 12, we present the complete confusion matrices for pseudo-labels.

## G INPUT SEGMENTATION

The images can be segmented according to the dominant feature activated at the pixel level within the input. In Figure 13, we highlight some of the segmentation outputs.

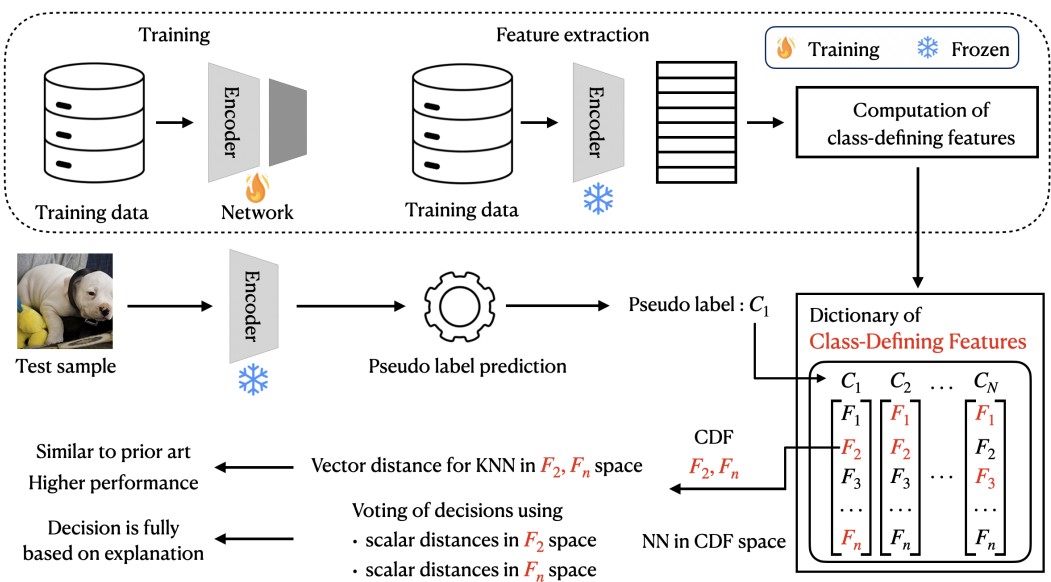

Figure 7: A method ablation for for sensitivity analysis, which uses the $\ell^2$ distances

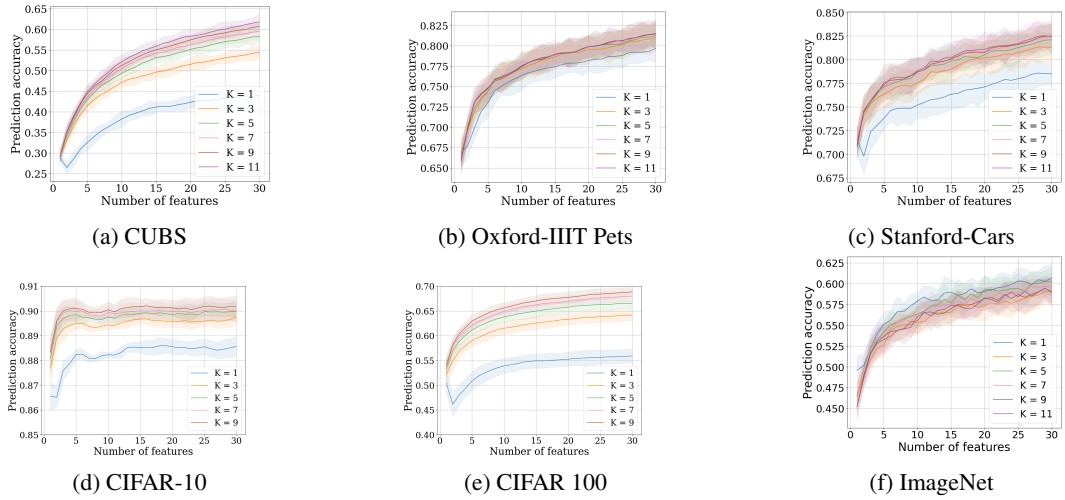

Figure 8: The performance of COMiX with different choices of hyperparameters. Each of the lines corresponds to a value of $K$ ($K$ nearest neighbours), and the horizontal axis shows the number of features $M$ used for prediction.

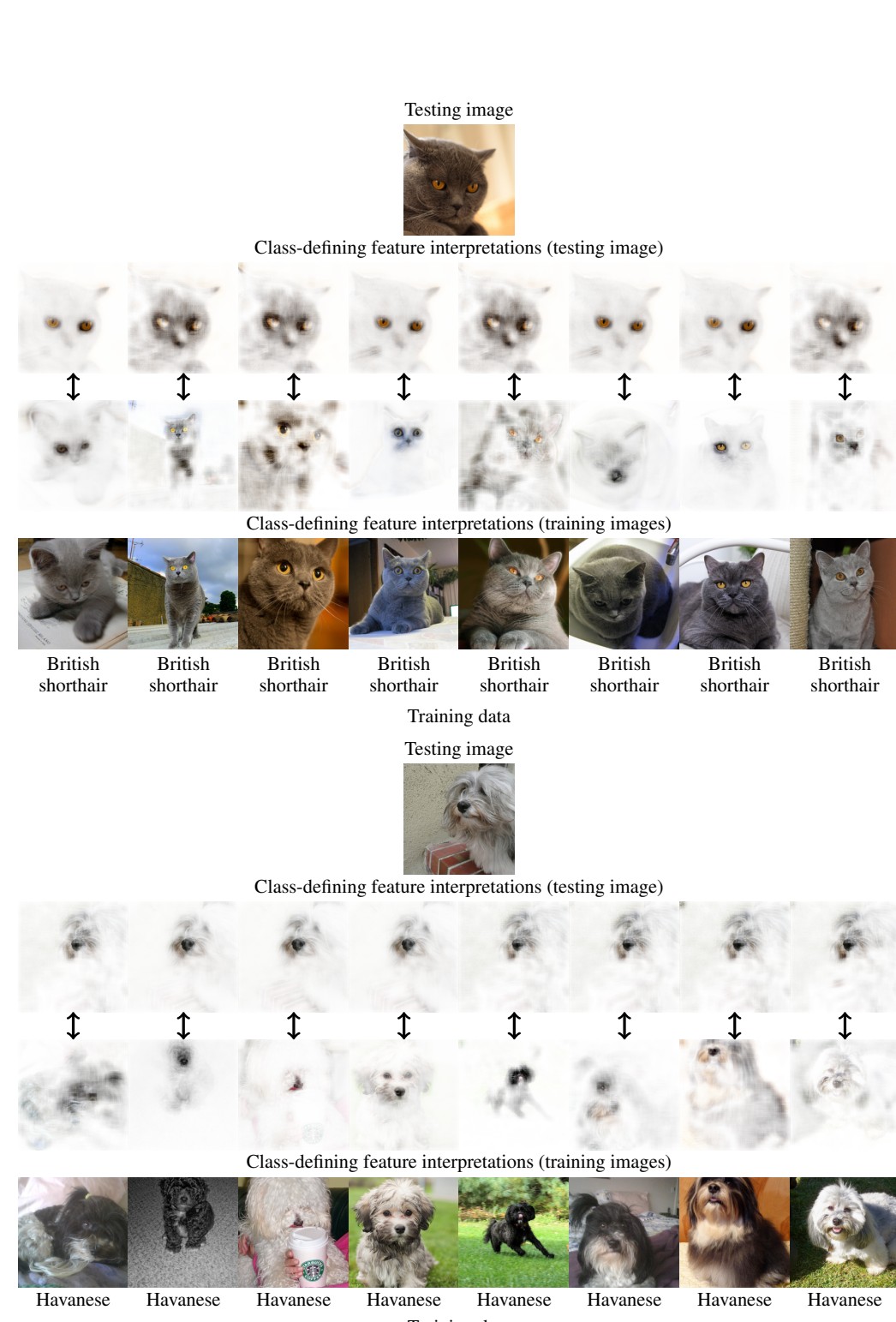

Figure 9: Additional qualitative results (Oxford-IIIT Pets)

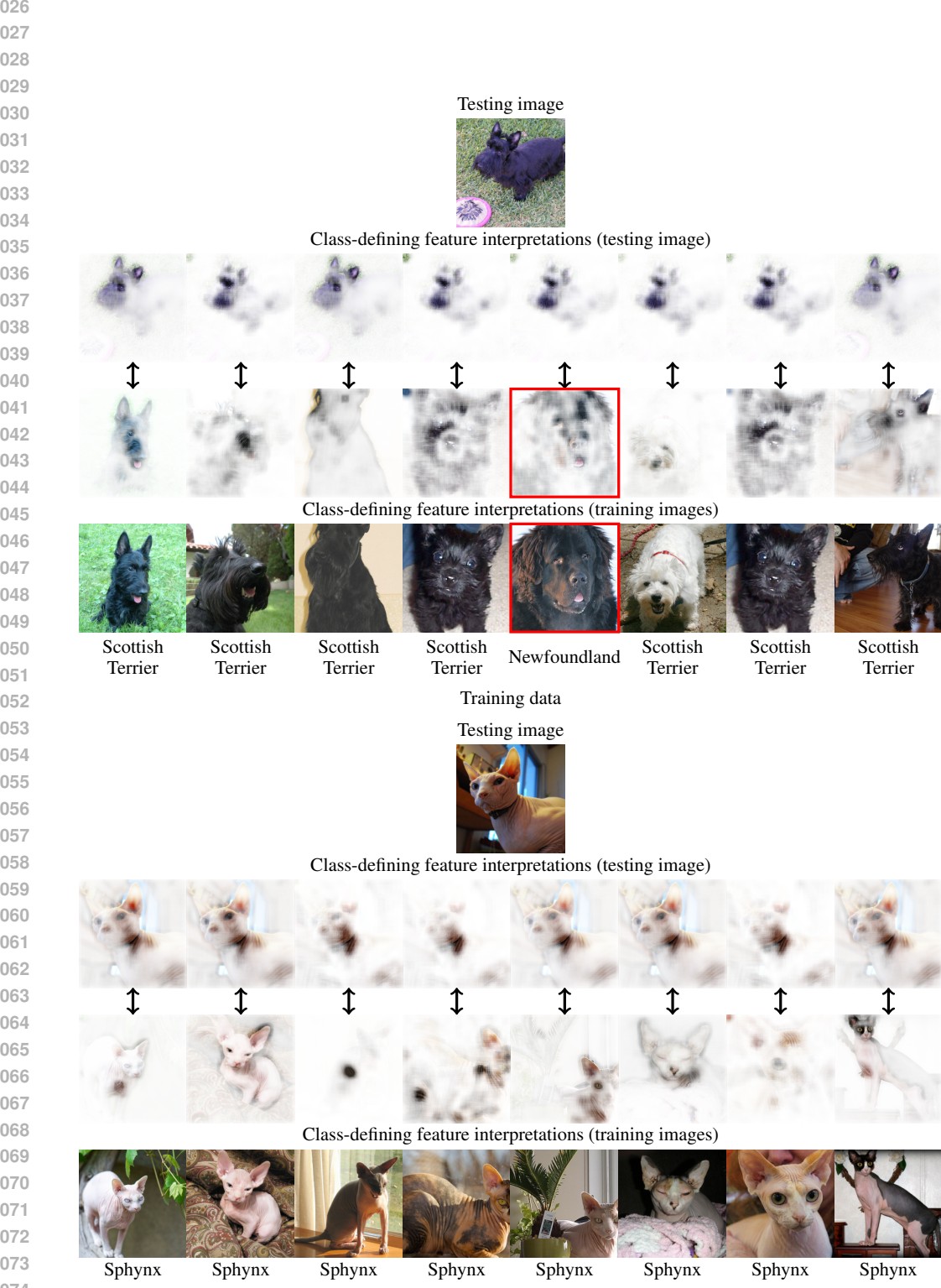

Figure 10: Additional qualitative results (Oxford-IIIT Pets)

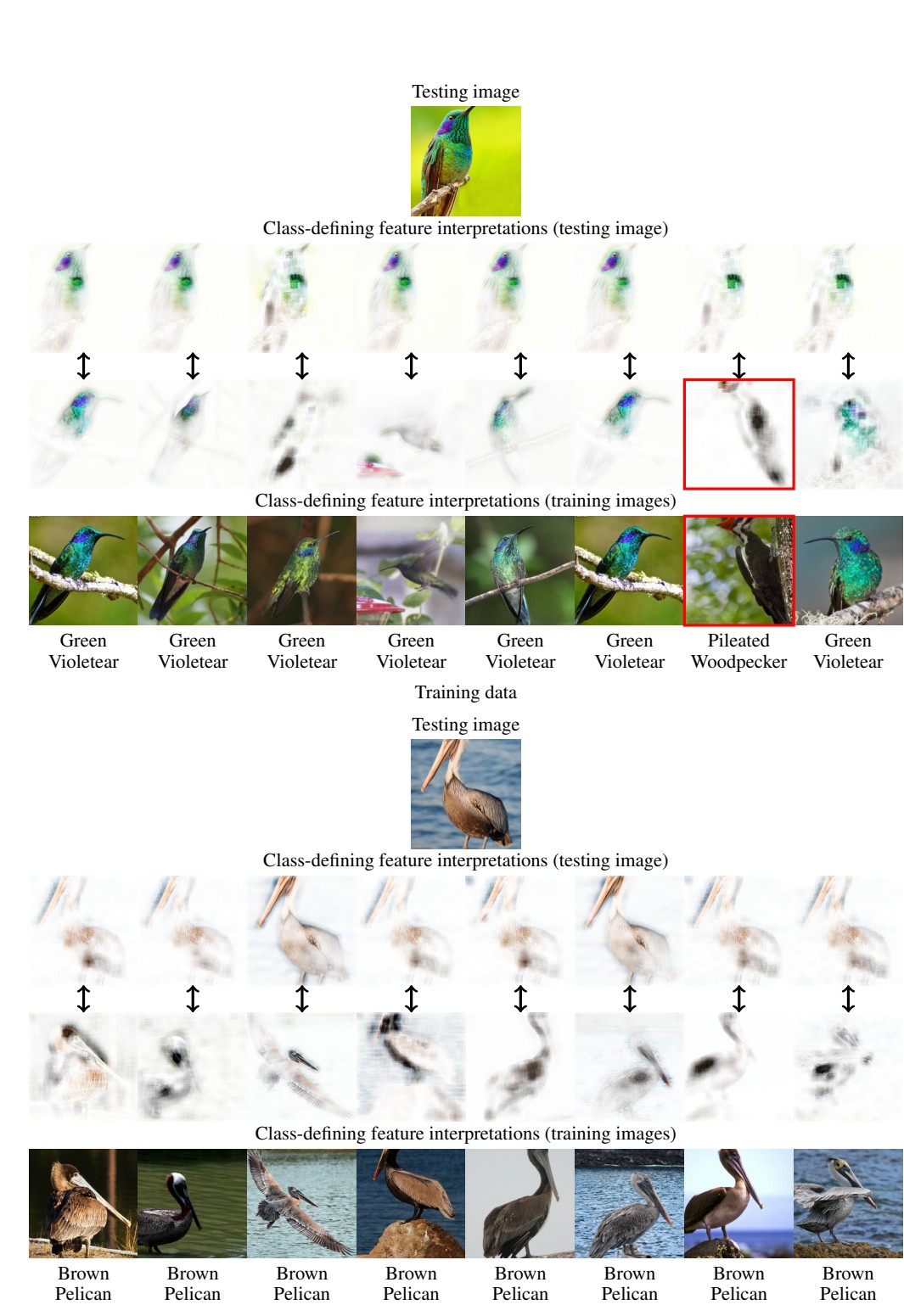

Figure 11: Additional qualitative results (CUB-200-2011)

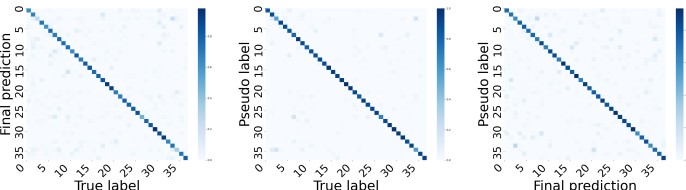

Figure 12: Confusion matrices for pseudo-labels: comparison between the ground-truth, final and pseudo-label on Oxford-IIIT Pets dataset

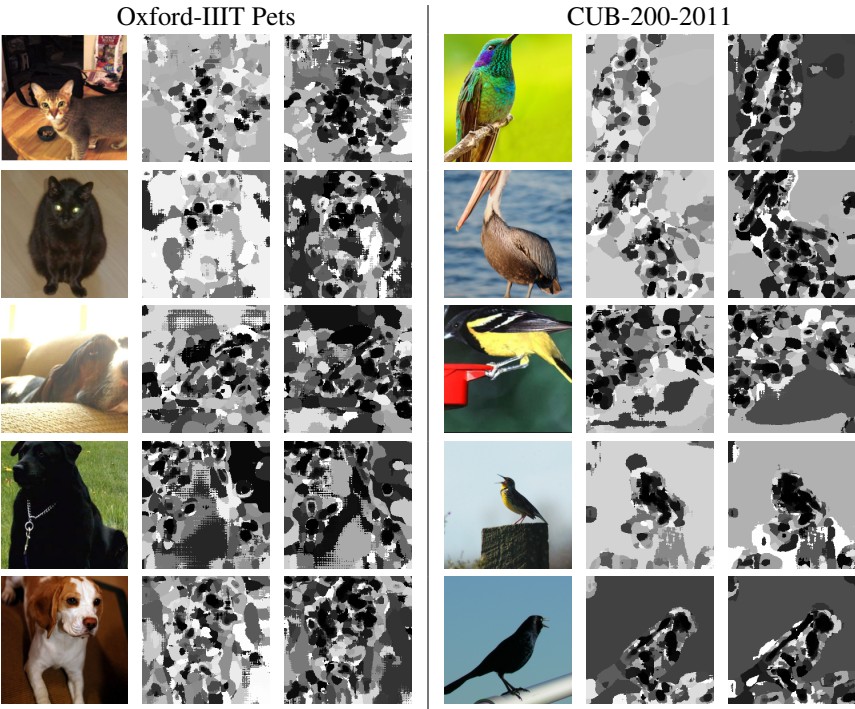

Figure 13: Image segmentation results: testing image, segmentations by leading CDFs, and segmentations by all features

## H    BROADER IMPACTS

While some of the existing *post hoc* explanation methods can explain the decision making, they do not follow the original decision making process Rudin (2019). This, therefore, cannot satisfy the current legal, ethical and policy-making needs. In contrast, by-design methods provide explanations which are causally linked with the decision making process. Such alternative is especially important for safety-critical applications, such as autonomous driving, robotics, medical imagery.

As finetuning-free learning was not considered the primary goal of this work, it was merely documented and not investigated further. It remains to be seen as to why COMiX results in surprisingly good finetuning-free performance.

## I    LIMITATIONS

Use of the pseudo-labels for preliminary selection of features can be also considered as a limitation, which is common for other works using concept-based interpretations due to the fact that feature selection necessitates pre-selection of the proposal class for subsequent refinement. Tan et al. (2024) describes the similar problem for their *post hoc* analysis method as a feature refinement problem.

