# OpenReview forum: "COMiX: Compositional explanations using prototypes"
_ICLR.cc/2025/Conference — ICLR 2025 Conference Withdrawn Submission_

### Official Review · Reviewer_at2g · 2024-10-22

**Soundness:** 1
**Presentation:** 2
**Contribution:** 1
**Rating:** 1
**Confidence:** 5

**Summary:**

The paper proposes a model that aims to be locally interpretable by design, as it combines B-cos Networks, which provide faithful saliency maps, with prototype methods. During inference, a pseudo-label is computed, whose M learned class features, or class prototypes, are then compared to the test image. The resulting classification seems to be the majority vote of the classes to which the most similar prototype belongs.
As B-cos Networks can provide faithful localizations on training and test images, this enables a visualization of the matching pixels.
As the features are not restricted to patches, the method is considered as object-level attention with prototypes.

**Strengths:**

The paper explores the important field of building interpretable models.

The core idea of the paper is presented clearly, with good figures 1 and 5.

Combining b-cos with prototypes is a novel combination.

The method is evaluated on a sufficient number of datasets and across various architectures.

**Weaknesses:**

The paper is missing the discussion of several competitors, e.g.:ProtoPool, Pip-Net, Q-SENN (citations at the bottom);
All of these focus on sparsity and Q-SENN and ProtoPool additionally aim for object-level prototypes or features, not restricting the prototypes to patches.
They further achieve better results in e.g accuracy and sparsity, which leads to global interpretability.
Notably, ProtoPool also uses exact training images as prototype for matching.
Thus, the novelty of the proposed method is very limited, as it just combines two existing methods, prototypes and b-cos, even without proper citation and comparison to in multiple ways superior SOTA. Notably, SOTA methods are backbone independent, so presumably compatible with b-cos.

The reported baselines are significantly too bad. Just training a vanilla resnet34 on CUB (on presumably the same data: 224x224, no crop) with hyperparameters optimized for a resnet50 on 448 gets 2.7 p.p. more accuracy.
This more than doubles the gap, and questions the validity of other results too.

The writing, especially of the method, is not clear:
 Eq. 4-11 lack explanations.
The algorithm 1 is unclear and lacks explanations.
Figure 2 is not clear.
What is trained when and how many settings (encoder training, CDF computation, inference?) exist and are shown is not clear.
The indices in eq.2-3 are unclear.
What is M set to?
Table 4 is not clear. What is the probed model for the other methods, and why is b-cos not a baseline?

Using a parametrized metric (PQ) with just one competitor is not convincing, when the parameters are chosen.
Additionally, the sparsity seems just slightly above a black-box model.

The definition of sufficiency in l308 seems different to the initially stated goal.

The standard deviations of results, excluding table 2,  and number of seeds per result are missing.

"Previous work has also shown that B-cos transformers inherently learn human-interpretable
features."  (l.217) needs a citation if available. I am not aware of any work showing that.


ProtoPool: Rymarczyk, Dawid, et al. "Interpretable image classification with differentiable prototypes assignment." European Conference on Computer Vision. Cham: Springer Nature Switzerland, 2022.
Pip-Net: Meike Nauta, Jörg Schlötterer, Maurice van Keulen, Christin Seifert (2023). “PIP-Net: Patch-Based Intuitive Prototypes for Interpretable Image Classification.” IEEE/CVF Conference on Computer Vision and Pattern Recognition (CVPR).
Q-SENN: Norrenbrock, Thomas, Marco Rudolph, and Bodo Rosenhahn. "Q-senn: Quantized self-explaining neural networks." Proceedings of the AAAI Conference on Artificial Intelligence. Vol. 38. No. 19. 2024.

**Questions:**

Am I misunderstanding the method?

Why is an uninterpretable pseudo-label necessary?

Did I use a wrong configuration for my baseline?

---

> ### Author Response · Authors · 2024-11-20
>
> > missing the discussion of several competitors.
>
> Many thanks, we will add the appropriate citations for the presented work into the related work section. SOTA methods are actually not backbone-independent. ProtoPNet, for example, relies upon the standard convolutional architecture, so not compatible with B-Cos (see [4] for the details). I understand that for the same exact reason the more recent works such as [5] do not provide non-convolutional backbones.
> In summary, we do not compete with ProtoPNet-style models, but propose a different type of  interpretable-AI model.  The goal of the paper was not to propose a better ProtoPNet or address its shortcomings, but to produce high-fidelity interpretations using both the training data and inherent explanations.
> To address this concern, we would then like to spell out the differences between the proposed method and the ProtoPNet family. We would argue (and address it in the text more clearly) that actually ProtoPNet-type models offer post hoc explanation in the following sense. They optimise latent representations of the prototypes (in case of ProtoConcept [1], for example, they optimise the parameters of the prototypical balls.  For these optimised latent representations, they either ‘project/push the testing example to the nearest training example‘ (ProtoPNet) or, in ProtoConcept, the authors visualise the training-data prototypes for the concepts contained within a ball. In contrast to this, the proposed method visualises the images which are directly forming a part of the decision-making process, thus providing fidelity of the explanations (see Steps 4 and 5 of Algorithm 1).
>
> Notably, contrary to the reviewer’s claim, ProtoPool does not use exact training images as prototypes but uses projection of learnt prototypes into the training images just like ProtoPNet. Page 7 of Rymarczyk et al (2022) states: “Prototypes projection is a step in the training process that allows prototypes visualization. It replaces each abstract prototype learned by the model with the representation of the nearest training patch” It means that although they use certain other innovations, their works’ contribution is different from ours and they do not use exact training images for matching but instead use prototype projection, similarly to the other methods in the ProtoPNet family.
> “even without proper citation” We would like to kindly clarify, which particular work, in the opinion of the reviewer, has not been properly cited while being used in this work. As we discuss above, we do not propose a better ProtoPNet, but to provide inherent prototypical interpretation for the models inherently interpretable through attribution maps, such as B-cos. As the authors, we do not know of any such work, and we do not think it is similar, in its aims or goals, to ProtoPNet. But if you have any work in mind we would gladly cite this work if the reviewer comes up with it.
> Neither does this method constitute a combination of the methods. (1) As we state above, we respectfully disagree that ProtoPool implements a method similar to the proposed one, and neither of the suggested methods implement decision making through actual images. Furthermore, our method searches for the similarities in the features for the whole image and does not select image patches, as it is done in ProtoPNet. Implicitly, the patch-dependence makes ProtoPNet family methods bound to the convolutional models. The proposed method does not have such a restriction. (2) Although we use B-cos backbone, it may be used with the other methods, which provide attribution maps as a part of the computational pipeline. It may also be converted from the existing models, using, for example, the results from B-cosification paper [3]
>
>
> > Reported baselines are significantly too bad.
>
> We would like to check where has this accuracy on ResNet34 been taken from and what is 2.7 p.p. more accuracy compared with ? We take our ResNet34 results, as stated in the paper, from Donnelly et al (2022), so it would be great if the reviewer could confirm the settings of their experiment.
>
> > Clarity of writing
>
> Many thanks, we are going to update the notation to improve the clarity. A detailed explanation of Eq 2n3 are given in appendix 1. This is directly taken from Bohle et al. Table 4 is generated using the same protocol as the same experiment protocol as prior art (mentioned in the main text). We add Bcos to the table for comparison in the revised text.

---

> > ### Author Response · Authors · 2024-11-20
> > **Continued**
> >
> > > Sparsity of the model
> >
> > The intention is not to show the performance against the competitor, but to show the notional PQ value comparing to the standard network. The comparison of sparsity in table is used to further motivate the use of B-cos encoder. Figure 6 shows the much higher sparsity of the proposed method. The performance of the method under as short as 12 features is reported. Also the method is sparse with respect to the number of train samples it uses for decision making (shown with the ablation with different K values). Further ablation is given in Figure 8 in the appendix.
> >
> > > Definition of sufficiency in l308 seems different to the initially stated goal.
> >
> > Many thanks for spotting this, in the next revision we will fix Line 96.
> > > The standard deviations
> >
> > It is due to the reason that the prior art did not report the standard deviations. We used five seeds in our experiments, and will add these details in the revised version.
> >
> > > Citation for B-cos transformers inherently learn human-interpretable features.
> >
> > We will modify the text to back up the claim. For the reference, the claim is presented in [1].
> >
> >
> > >Question 1:
> >
> > In summary, we think there are multiple differences between the proposed method and the ProtoPNet family as outlined above:
> > The proposed method is not a post-hoc explanation method, that explains a decision, but a decision making mechanism that has a ‘this looks like that’ style human understandable step in the decision making.
> > The interpretation, by using a B-cos encoder, completely explains the decision making, truthfully (without approximations).
> > The proposed method supports a range of architectures including the attention-based ones, while ProtoPNet-based methods are designed for the CNNs.
> > The proposed method uses directly the training data for the decision making thus improving fidelity; the ProtoPNet-based method do not
> > This method does not use patch extraction, but the ability of inherently explainable methods such as B-cos to generate saliency maps. In the other words, we don’t use patches but the whole images.
> >
> > >Question 2:
> >
> > We present our decision making through a set of training examples. Humans explain the decisions of different classes differently. For instance "What makes a cat a cat" is different from "what makes a car a car". We wanted to include this specificity to align to human explanation in the method. We could compute global feature bank for explanations. Following the reviewer comment we show the robustness of the proposed framework to choosing other class features to make predictions. Results are added to the appendix.
> > This, in a way, is a chicken-and-egg problem: we need to select class-specific features, but to do this, we need to know the class label. For that purpose, we need pseudo-labels.
> > On the lack of interpretability of the pseudo-label, the idea is that the pseudo label has minimal impact on the decision making (see Figure 11). We derive the decision from the training examples and the selected features.
> >
> > >Question 3:
> >
> > We are not entirely sure which baseline this question refers to, but we hope that the outlined differences between the proposed method and ProtoPNet family methods would clarify the rationale behind the experimental setting. Furthermore, in Table 3, we present a comparison with a number of ProtoPNet-based methods. It happens, however, that because they rely upon selection of (spatial) patches and as they do not work for the transformer architecture, we cannot present the like-for-like comparison with such methods.
> >
> > [1] B-Cos Aligned Transformers Learn Human-Interpretable Features. Tran et al (International Conference on Medical Image Computing and Computer-Assisted Intervention)
> >
> > [2] ProtoPool: Rymarczyk, Dawid, et al. "Interpretable image classification with differentiable prototypes assignment." European Conference on Computer Vision.
> >
> > [3]  Arya et al (2024) B-cosification: Transforming Deep Neural Networks to be Inherently Interpretable
> >
> > [4] Xue et al (2022) ProtoPFormer: Concentrating on Prototypical Parts in Vision Transformers for Interpretable Image Recognition
> >
> > [5] Ma, Chiyu, et al. "This looks like those: Illuminating prototypical concepts using multiple visualizations." Advances in Neural Information Processing Systems 36 (2024).

---

> > > ### Comment · Reviewer_at2g · 2024-11-21
> > > **Official Comment by Reviewer at2g**
> > >
> > > > ProtoPNet, for example, relies upon the standard convolutional architecture, so not compatible with B-Cos (see [4] for the details)
> > >
> > > This claim is not clear, as in this very paper, they use ProtoPNet with ViTs. In [4] they apply ProtoPNet to Vits and discuss reasons why it does not work as well, but it is still compatible. When looking at the Bcos-Vits (https://github.com/B-cos/B-cos-v2), it does seem like bcos-vits show a larger gap compared to the non-bcos versions too.   So the entire motivation why this method is forced to use Bcos-ViTs and why it therefore compares with no competitors is not clear.  Especially, why would ProtoPool or ProtoPNet not be compatible with a B-cos CNN?
> > >
> > > Without proper citation refers to no citation for the works that are most similar, especially ProtoPool.
> > > I am also still not sure if I get the novelty compared to Protopool. As you cited, Protopool replaces prototypes with the representations of training patches. Thus, it exactly computes the similarity in its representation between a test image and a training image. Is the main difference from an interpretability point in patches as opposed to entire training images?
> > >
> > >
> > > > We will modify the text to back up the claim. For the reference, the claim is presented in [1].
> > >
> > > Well, while the title of [1] says that, it is not actually evaluated in the paper. Their localization is just more focussed on the relevant parts as assigned by humans, so preferable to a non-bcos transformer. More fine-grained localization does not necessarily indicate human like features. In that paper, e.g. cells could have a different texture, that a human does not perceive.
> > >
> > > > We would like to check where has this accuracy on ResNet34 been taken from and what is 2.7 p.p. more accuracy compared with ? We take our ResNet34 results, as stated in the paper, from Donnelly et al (2022), so it would be great if the reviewer could confirm the settings of their experiment.
> > >
> > > I repeated it with this public implementation: https://github.com/ThomasNorr/Q-SENN, reaching 78.5 across 3 seeds.  As stated in my initial comment, image size is 224, no crop is used and no additional loss.  The baseline results cited just seem way too low.

---

> ### Author Response · Authors · 2024-11-21
>
> Thank you for a prompt response and for such a thorough discussion about the concerns.
>
> > “This claim is not clear, as in this very paper, they use ProtoPNet with ViTs. In [4] they apply ProtoPNet to Vits and discuss reasons why it does not work as well, but it is still compatible. When looking at the Bcos-Vits (https://github.com/B-cos/B-cos-v2), it does seem like bcos-vits show a larger gap compared to the non-bcos versions too. So the entire motivation why this method is forced to use Bcos-ViTs and why it therefore compares with no competitors is not clear. Especially, why would ProtoPool or ProtoPNet not be compatible with a B-cos CNN?" Without proper citation refers to no citation for the works that are most similar, especially ProtoPool. I am also still not sure if I get the novelty compared to Protopool. As you cited, Protopool replaces prototypes with the representations of training patches. Thus, it exactly computes the similarity in its representation between a test image and a training image. Is the main difference from an interpretability point in patches as opposed to entire training images?”
>
> We will try and clarify upon the difference between our work and ProtoPNet-style models. In short, we neither see our work similar to ProtoPNet beyond the similarity in the presentation of explanation, as both architecture and the problem is different. In summary, we believe that ProtoPNet family is just a different family of models, which is only related by the means of ptresentation of explanations through training data. The activation maps in ProtoPNet represent the similarity with a patch (i.e., it is a function of a patch and an input), while in our method we present the attribution for a particular feature (i.e., it is a function of an input and the feature id, not a similarity) so they are not like-for-like.
> The differences between the ProtoPNet models and the proposed model as per the ProtoPNet paper:
>
> - ProtoPNet obtains the activation maps using the following procedure : This activation map preserves the spatial relation of the convolutional output, and can be upsampled to the size of the input image to produce a heat map that identifies which part of the input image is most similar to the learned prototype. We take advantage of self-interpretable architectures such as B-cos, which can produce attribution maps for any layer and any feature by-design, directly of a size of an input image. Hence, our method’s attribution maps are not related to the prototypes and only to the input, while the ProtoPNet’s activation maps are related to the similarity with learnable prototypes.
>
> - There is a difference between the interpretation of the ProtoPNet’s activation maps and the COMiX’s attribution maps. ProtoPNet maps represent the highest activation for a particular prototype. COMiX’s maps represent the activation for a neural-network feature (and are thus only dependent upon the input data).
>
> - ProtoPNet learns the set of per-patch prototypes (from ProtoPNet paper, Section 2.2: “we jointly optimize the convolutional layers’ parameters wconv and the prototypes P“). We do not do this, we  attribute the decisions to the whole set of training data. We are motivated by the fact that we need to connect the decision with the training data.
>
> - ProtoPNet does not connect the decisions directly with the training data, it connects it with the synthetic prototypes, learnt from the training data. It is a substantial difference: the prototypes in ProtoPNet are optimised with the training data matches found post hoc, while we directly express the decision through the training data. ProtoPNet uses the following technique for visualisation: “To be able to visualize the prototypes as training image patches, we project (“push”) each prototype pj onto the nearest latent training patch from the same class as that of pj . In this way, we can conceptually equate each prototype with a training image patch” It means that these nearest latent training patches in ProtoPNet only approximate the prototypes. Thanks to the by-design explanation provided by B-cos family models, we avoid such approximation.

---

> ### Author Response · Authors · 2024-11-21
> **continued**
>
> In a more targeted way, let us look into the claim that ProtoPool may be the most similar method.
>
> - Although the mechanism to obtain the activation maps is different from ProtoPNet, ProtoPool also produces prototype activation maps which reflect the similarity between the prototype and the input
>
>  -  ProtoPool also uses the projection of the prototypes to the training image patches
>
>  - ProtoPool method is also focused on selection of learnable prototypes, and improves upon ProtoPNet
>
>  - ProtoPool does not produce per-feature explanation of decision making. Instead, it explains the whole decision through similarity with prototypes. We produce interpretations for every class-defining feature.
>
>  Therefore, we believe that ProtoPool is an entirely different method solving a different problem, addressing the architectural shortcomings of ProtoPNet such as per-class prototypes amongst others, which is the reason why we don’t present our work as an improvement upon ProtoPool. We will add the citation of ProtoPool into the review of interpretable methods.  In terms of similar works within the ProtoPNet family, we would think Ma et al [5] is actually a closer match, as suggested and discussed with Reviewer HwsD.

---

> > ### Author Response · Authors · 2024-11-21
> >
> > In a sense, we think these are all the compelling reasons to state that ProtoPNet (and its variations) is a different algorithm addressing different problems. They solve the problem of interpretation through learnable prototypes, we solve the problem of taking advantage of inherently interpretable methods with input-data attribution, such as B-Cos, to produce interpretation with respect to the training data.
> >
> > >“This claim is not clear, as in this very paper, they use ProtoPNet with ViTs.”
> >
> > The first aspect is that they do indeed use the version of ProtoPNet with ViTs, but they propose a modified version for ViTs to improve its efficiency and make explanations meaningful. Then, the quality of explanations, as outlined in Figure 4 of [4], is not really satisfactory for the vanilla ProtoPNet. Finally, the direct comparison with ProtoPNet on the attribution maps would not be correct, as the proposed method’s attribution maps are conceptually different from ProtoPNet’s (attribution to the input vs similarity, as discussed above).
> >
> > >“So the entire motivation why this method is forced to use Bcos-ViTs and why it therefore compares with no competitors is not clear. “
> >
> > First, we do compare our model with ProtoPNet in Table 3, in a way similar to Donnelly et al (2023). Is there any other comparison that the reviewer has in mind? Second, the entire motivation of using Bcos-ViTs is to take advantage of their per-image attribution maps in a way that is faithful to the computation.  ProtoPNet does not generate per-image attribution maps, but produces instead activations for learnt prototypes. In our opinion, it is a different activation map (it depends upon the prototypes, whereas Bcos-ViT’s attribution map does not). We are not aware of any other method that achieves that goal but it may be possible to use the other similar methods to Bcos.  Without inherently explainable attributions it would not provide fidelity, and it cannot be directly compared with ProtoPNet explanation in a qualitative way, because their activation maps represent fundamentally different quantity.
> >
> > >“Their localization is just more focussed on the relevant parts as assigned by humans, so preferable to a non-bcos transformer. “
> >
> > Yes, so it is consistent with our claim that B-cos transformers inherently (see for proofs of ‘inherently’ in Böhle et al, 2024) learn (by design of Böhle et al, 2024) human-interpretable features (see [1], based on the description of Figure 1 in the appendix, claiming : “In a blinded study, a domain expert ranked models (lower is better) based on whether the models focus on biomedically relevant features that are known in the literature to be important for diagnosis.“).
> > Böhle, Moritz, et al. "B-cos Alignment for Inherently Interpretable CNNs and Vision Transformers." IEEE Transactions on Pattern Analysis and Machine Intelligence (2024).
> >
> > >“I repeated it with this public implementation: https://github.com/ThomasNorr/Q-SENN, reaching 78.5 across 3 seeds. As stated in my initial comment, image size is 224, no crop is used and no additional loss. The baseline results cited just seem way too low.”
> >
> > One of the possible reasons might be that the baseline, published by Donnelly et al (2022) uses the ImageNet pretraining, also we don’t think the hyper parameters are selected using ResNet50. As a peer-reviewed paper, Donelly et al (2022) constitutes an important baseline recognised by the community. In our case, similar to Donelly et al (2022), it would be also fair to use ImageNet as we use ImageNet pretraining in our scenarios.

---

### Official Review · Reviewer_izKH · 2024-10-30

**Soundness:** 3
**Presentation:** 3
**Contribution:** 2
**Rating:** 6
**Confidence:** 3

**Summary:**

Being able to interpret deep neural networks' decisions is becoming increasingly important with the integration of AI into high-risk areas such as autonomous driving and medical diagnosis. Existing methods attempt to explain existing model post-hoc, but may not be faithful to the inner workings of the network. The authors present COMiX, a prototype based method for image classification that links the prediction logic to image parts in the training data. COMiX utilizes B-cos networks for transparency (due to its linearity) from the input pixels to the class defining features used for classification. COMiX then aligns the features from the test image to the features in the training images to perform a KNN-based prediction in order to establish a “this part looks like this one” example-based prediction. The authors measure their model’s ability on four desiderata: fidelity, sparsity, necessity, and sufficiency. They apply their method to several vision backbones and one several popular image datasets. In addition, they qualitatively show the effectiveness of their method for explaining its predictions.

**Strengths:**

- Qualitative results are strong and convincing
- Modeling / approach is simple
- Robust use in different models / architectures
- Robust ability across datasets
- Creative use with B-cos networks and label aggregation
- Sufficient set of quantitative evaluation metrics
- Hyperparameter analysis / ablation studies present
- Better sparsity than ViT baseline and best insertion scores w/ competitive deletion scores.

**Weaknesses:**

- The interpretability framework novelty isn't significantly more compared to ProtoPNet (Seems like another ‘this looks like that’ explanation just reframed)
  - Much of the framework is similar to ProtoPNet with the exception of using pretrained features as concepts (as opposed to specialized vectors), a b-cos backbone, and KNN based prediction on feature similarity.

- Presentation is unclear at times:
  - Motivation/need for sufficiency is unclear
  - Notation seems convoluted
    - Confused about $s_{L}(x;\theta)$ It seems that it should replace $W_{1->L}(x;\theta)$ in equation 4.
  - Not entirely clear on the attribution method. Is it which feature a pixel contributes most to?

**Questions:**

- Could you clear up my understanding of the attribution used to make the visualization?
- What is $s_L(x:\theta)$ exactly? Could you provide a better/clearer definition?
- I’m not entirely sure of the definition of necessity, and I’m unsure of the need for sufficiency. It seems redundant. Could the authors clarify necessity, and the need for sufficiency as a desiderata?
- What is the speed of this method compared to others, given it has to look through the entire training data?

---

> ### Author Response · Authors · 2024-11-20
>
> First of all, we would like to thank the reviewer for their constructive feedback.
>
> > novelty isn't significantly
>
> We would think that there are even more related methods from the ProtoPNet family. The more recent and more relevant paper, as also suggested by Reviewer HwsD,  would be Ma et al (2024) (ProtoConcept), which performs visualisations for multiple image features in a prototypical setting.
> However, even this paper is not implementing the interpretation the same way. Their idea is to perform the explanations through multiple image patches. In the centre of our idea is the fact that inherently interpretable models, such as B-cos, can explain the decisions in different ways by every feature. In this way, our method bridges the gap between the concept bottleneck and the prototypical models.
>
> We would then like to spell out the differences between the proposed method and the ProtoPNet family. We would argue (and address it in the text more clearly) that actually ProtoPNet-type models offer post hoc explanation in the following sense. They optimise latent representations of the prototypes (in case of ProtoConcept [1], for example, they optimise the parameters of the prototypical balls.  For these optimised latent representations, they either ‘project/push the training example to the nearest training example‘ (ProtoPNet) or, in ProtoConcept, the authors visualise the training-data prototypes for the concepts contained within a ball. In contrast to this, the proposed method visualises the images which are directly forming a part of the decision-making process, thus providing fidelity of the explanations (Steps 4 and 5 of Algorithm 1).
> Furthermore, ProtoPNet-type models rely on selection of patches. It creates inherent bias to spatial proximity, and makes the model implicitly dependent upon the convolutional architecture as opposed to the alternatives such as transformers (see [2] for details why the ProtoPNet-type models cannot be used without modifications with the transformer architecture). The purpose of the paper has not been to provide an improved ProtoPNet architecture but to provide inherent interpretations in the style of ProtopNet explanation (this looks like that), without any post hoc analysis, through prototypes and using intrinsically interpretable models producing saliency maps, such as B-cos. We will update the text accordingly to provide better clarity upon this point, as it seems to be the common point between the reviewers.
>
>
>
> > Motivation/need for sufficiency is unclear
>
> The motivation for sufficiency is highlighted by our goal to provide high-fidelity explanations and demonstrate that we achieve the goal of meeting the well-established desiderata for interpretable ML. To address it, in the next revision we will fix Line 96 to make it synchronised with the formalisation of sufficiency in Line 308. We believe it is standard for the interpretable ML literature (see, e.g. [3] which is defining sufficiency metrics for quantifying faithfulness. We will add further references highlighting the role of sufficiency in the related work.
>
> > Notation seems convoluted
>
> The $s^i_{L}(x;\theta)$ are the per-feature components of $W_{1->L}(x;\theta)$, which we introduce to make Equation 6 more readable. We will refine the notation accordingly to improve the presentation.
>
> > Not entirely clear on the attribution method. Is it which feature a pixel contributes most to?
>
> Many thanks for this, you are absolutely right on this. We will update the notation accordingly. The attribution is obtained by a truthful explanation of the encoding of the features using  B-Cos models to provide the per-pixel contributions by-design  (see Section 3.1 and Appendix A for more detail). We present the per-pixel attribution maps for a particular feature. See the answer to the question below on how exactly that map is produced.

---

> > ### Author Response · Authors · 2024-11-20
> > **Continued**
> >
> > Question1: The attribution maps come from the B-cos model and are not provided post hoc. To be more precise, in line 218 we mention that ‘We compute the explanation for a feature i in the L-th layer as $W_{1-> L} (\mathbf{x}, \theta)^i$’. It means that we use the direct visualisation of (input-dependent) B-cos weights for a particular layer as opposed to a post hoc visualisation. We use colour coding of these weights as per Section 4 of [4].
> >
> > Question 2: The $s^i_{L}(x;\theta)$ are the per-feature components of $W_{1->L}(x;\theta)$, which we introduce to make Equation 6 more readable. We will refine the notation accordingly to improve the presentation.
> >
> > Question3: The main reason behind providing these results were to emphasise the fact that we do not use post hoc analysis and we improve the fidelity of the interpretations. As a part of it, we want the interpretations to be necessary (i.e., presence of the elements of the explanation should be necessary for the decision making) and sufficient (the same explanation implies the same output). Such an approach is actually commonplace in the xAI literature, e.g. in [5] amongst others they present the following argument:
> > “necessity ensures that a feature or condition is indispensable for an outcome—removing it would prevent the model's decision, while sufficiency guarantees that its presence alone can produce the outcome. A condition is necessary if its absence alters the outcome, indicating its indispensable role. Conversely, a condition is sufficient if its presence alone can produce the outcome, highlighting its standalone explanatory power. “ Various prior explanation methods have used these complementary measures to show the efficacy of the proposed explanation method.
> >
> >
> > Question 4: We compute the features of the train data and then the CDF as part of the training process. As you pointed out the method indeed involves comparing with the entire train data features at inference. But we do not perform a dataset subsampling to ensure the explanation indeed reflects the entire data the encoder was trained on. The claims of the paper, line 74, states: “This approach goes beyond attribution map predictions and presents a model, by design, that traces the decision to the original training data. ”. We also say in the paper, line 135: ‘in addition, COMiX traces these concepts back to the training data and provides inherent, by-design, interpretations.  ’An error in prediction when explained can help spot the faulty data points in the train data if there was any.
> >
> > [1] Ma, Chiyu, et al. "This looks like those: Illuminating prototypical concepts using multiple visualizations." Advances in Neural Information Processing Systems 36 (2024).
> > [2]  Xue et al (2022) ProtoPFormer: Concentrating on Prototypical Parts in Vision Transformers for Interpretable Image Recognition
> > [3] Dasgupta et al {2022} Framework for Evaluating Faithfulness of Local Explanations, ICML 2022
> > [4] Bohle et al (2022) B-cos Networks: Alignment is All We Need for Interpretability, CVPR 2022
> > [5] Watson et al (2021)  Local Explanations via Necessity and Sufficiency: Unifying Theory and Practice

---

> > > ### Comment · Reviewer_izKH · 2024-11-25
> > >
> > > Thank you for clearing up my confusions and answering my questions! While you're responses were sufficient for my points, I'm still assessing comments and responses from other reviewers. At this time, my score will remain the same.

---

### Official Review · Reviewer_zkRt · 2024-11-03

**Soundness:** 2
**Presentation:** 1
**Contribution:** 2
**Rating:** 3
**Confidence:** 3

**Summary:**

In this paper, the authors proposed COMiX, a method for constructing an interpretable image classifier from a trained B-cos network. In particular, the method starts by training a base B-cos network, and finding M class-defining B-cos features for each class. For a given input image, COMiX first computes a transformation matrix corresponding to the composition of all the layers in the base B-cos network, and then selects a pseudo-label based on the closest training image in terms of the B-cos network's output. For each class-defining feature belonging to the pseudo-label class, the method computes a class prediction based on K nearest training images in terms of that feature's values, and the final prediction is made using majority voting. The authors compared their COMiX classifiers with competing methods (e.g., baseline convolutional networks, ProtoPNet models, B-cos networks) on a number of datasets, and found that their COMiX classifiers performed similarly (or better) to other methods in terms of accuracy and interpretability metrics.

**Strengths:**

- Originality: The paper proposed an elaborate scheme to turn a B-cos network into a model with a capability to perform case-based reasoning (using k-nearest neighbors).
- Quality: The proposed method did not significantly degrade the classification performance.
- Clarity: The introduction is well-written, and the paper is well-motivated.
- Significance: Interpretability is an important topic.

**Weaknesses:**

- Originality: The proposed form of interpretability ("this part of the test image looks like that part of a training image") has been explored in prior work (e.g., ProtoPNet). There is no novelty here.
- Quality: The proposed method constructs a model that is not trainable end-to-end. Also, the proposed method is biased toward the pseudo-label predicted by the B-cos network, since the selection of class-defining features are based on the predicted pseudo-label.
- Quality: The accuracy of COMiX is not particularly strong.
- Clarity: The section describing the algorithm of COMiX is difficult to follow. The presentation is not clear. The notations are confusing. In particular, how is the class-defining features selected using equations (7) and (8), and how is the mutual information computed (equation (8))?
- Significance: Given that there is little innovation in terms of interpretability and the accuracy results are mediocre, the proposed method cannot significantly advance the field of interpretable machine learning.

**Questions:**

How is the class-defining features selected using equations (7) and (8)?
How is the mutual information computed using equation (8)?

**Details Of Ethics Concerns:**

N/A.

---

> ### Author Response · Authors · 2024-11-20
>
> We thank the reviewer for the comments and constructive feedback. Below we present the clarifications on the weakness and answer the questions.
>
>
> Originality: We would suggest that more recent and more relevant paper, as also suggested by Reviewer HwsD,  would be Ma et al (2024) (ProtoConcept), which performs visualisations for multiple image features in a prototypical setting. However, even this paper is not solving the problem the same way. Their idea is to perform the explanations through multiple image patches. In the centre of our idea is the fact that inherently interpretable models, such as B-cos, can explain the decisions in different ways by every feature. In this way, our method bridges the gap between the concept bottleneck and the prototypical models, while, in contrast to ProtoConcept, avoiding altogether to explain it post hoc and instead providing interpretation at the same time as the decision. Below, we explain the last point in more detail, spelling out the differences between the proposed method and the ProtoPNet family.
>
> We would argue (and we will address it in the text more clearly) that actually ProtoPNet-type models offer post hoc explanation in the following sense. They optimise latent representations of the prototypes (in case of ProtoConcept [1], for example, they optimise the parameters of the prototypical balls.  For these optimised latent representations, they either ‘project/push the training example to the nearest training example‘ (ProtoPNet) or, in ProtoConcept, the authors visualise the training-data prototypes for the concepts contained within a ball. In contrast to this, the proposed method visualises the images which are directly forming a part of the decision-making process, thus providing fidelity of the explanations (see Steps 4 and 5 of Algorithm 1).
> We will update the text accordingly to provide better clarity upon this point, as it seems to be the common point between the reviewers.
>
> [1] Ma, Chiyu, et al. "This looks like those: Illuminating prototypical concepts using multiple visualizations." Advances in Neural Information Processing Systems 36 (2024).
>
> Quality: First of all, we would question why a model not being trainable end-to-end is a disadvantage per se. The selection of features and prototypes is a common problem between the concept- and prototype-based methods as the explanation over the whole feature set (e.g.., K nearest neighbours and 1024 features) is not viable. Nauta et al (2023), for example, dedicate Section 6.2 to describing the process of selection for the explanation size in the  prototypical model  setting.
> And to reiterate the point above, the principal difference between our setting and the ProtoPNet family of the models  is that they find the training examples post hoc while we present our decision making through a set of training examples. This necessitates  using only a handful of features. This, in a way, is a chicken-and-egg problem: we need to select class-specific values, but to do this, we need to know the class label. For that purpose, we need pseudo-labels to select the features.
>
> Quality: We use a B-cos network as the backbone and propose an inference method so that the explanation is part of the inference mechanism, i.e. the explanation is not post hoc.
> Instead of explaining the decision that is taken post-hoc through prototypes as it is done in ProtoPNet-type models, the decision making mechanism happens through a ‘this looks like that’-human understandable bottleneck. The trade-off between the inherent prototypical explanations on one hand, and the accuracy on the other hand is therefore the reason behind the inferior performance.
>
> Clarity:  Many thanks for helping us improve the presentation of the paper. We will update the notation accordingly.
>
> Significance: We think this stems from the suggestion that the method is, in fact, a variation of ProtoPNet. We do not think so: first, as we discussed above, ProtoPNet is not even the closest method out of the ProtoPNet family. Second, the principal contribution is the prototypical method which explicitly derives the decision from the training data, as opposed to post hoc assimilation of the similar training data to provide the explanation for the decisions.
>
> Question: Many thanks for this comment, at the moment we are updating the notation to provide better clarity.

---

> > ### Comment · Reviewer_zkRt · 2024-11-25
> > **Thank you for your response**
> >
> > After reading the response and other reviewers' comments, I am still concerned with the clarity of the paper. I still don't see additional clarifications on equations (7) and (8), regarding how the mutual information maximization is computed. The lack of baseline comparisons with other competing methods (as raised by Reviewer HwsD) is another concern. This should be done even if you argue that ProtoPNet-type models differ from your formulation.
> >
> > Considering the current state of the paper, I will keep my original rating.

---

### Official Review · Reviewer_HwsD · 2024-11-04

**Soundness:** 2
**Presentation:** 2
**Contribution:** 2
**Rating:** 3
**Confidence:** 3

**Summary:**

The authors introduced COMiX, an intrinsic explainable artificial intelligence (XAI) method designed to accurately identify prototypical regions within a test image and correlate them with corresponding regions in training images. This method is based on the feature encoder of the trained B-cos network and emphasizes the extraction of class-defining features that serve as prototypes, which are subsequently utilized for class predictions.

**Strengths:**

- S1: The proposed method is well-motivated, and the relationship between the prototypes and the class-defining features is clear. The requirements for intrinsic interpretability that the authors address are essential components of intrinsic XAI approaches.

**Weaknesses:**

- W1: One of my primary concerns pertains to the insufficient details regarding the essential computations involved in the proposed method, which necessitate considerable computational resources. In particular, additional clarification is needed on the computation of mutual information maximization as described in Equation 8. Specifically, how are p(F_j) and p(l(F_j) = c) calculated? Does this process necessitate traversing every row of W_{1 \to L}(d, \theta)? This indeed involves significant computational effort, alongside other tasks requiring substantial computation, such as the generation of pseudolabels for each CDF.

- W2: A secondary concern is the absence of baseline comparisons within the experimental results. While the authors sought to relate their method to prototypical approaches, they did not include comparisons with other state-of-the-art prototype-based methods, such as PIP-Net [1] and ProtoConcept [2], which demonstrate superior performance in accuracy evaluations. Additionally, Table 3 would benefit from including more baseline comparisons, as suggested by references [2, 4].

- W3: In Table 2, the proposed method demonstrates inferior performance relative to the original B-cos in multiple cases, raising concerns about the suggested approach's efficacy. Additionally, it would be valuable to explore the performance of the end-to-end model by training the feature extractor of the B-cos.

- W4: The experiment regarding sparsity requires further validation. Most studies on prototypical learning have addressed the impact of increased sparsity by utilizing global and local explanations or by adjusting the number of prototypes employed [1, 2]. Although Figure 6 presents an ablation study on the size of K, it would be beneficial to enhance this section by incorporating additional baseline comparisons.

- W5: The experiment focusing on prototypical explanations was conducted solely in a qualitative way. For a quantitative approach, please refer to [2]. Additionally, refer to references [3] and [4] for insights into unsupervised concept discovery.

- Reference
- [1] Nauta, Meike, et al. "Pip-net: Patch-based intuitive prototypes for interpretable image classification." Proceedings of the IEEE/CVF Conference on Computer Vision and Pattern Recognition. 2023.
- [2] Ma, Chiyu, et al. "This looks like those: Illuminating prototypical concepts using multiple visualizations." Advances in Neural Information Processing Systems 36 (2024).
- [3] Wang, Bowen, et al. "Learning bottleneck concepts in image classification." Proceedings of the ieee/cvf conference on computer vision and pattern recognition. 2023.
- [4] Hong, Jinyung,  Keun Hee Park, and Theodore P. Pavlic. "Concept-centric transformers:  Enhancing model interpretability through object-centric concept learning within a shared global workspace." Proceedings of the IEEE/CVF Winter Conference on Applications of Computer Vision. 2024.

**Questions:**

Most of my main concerns are listed in the Weakness section. Here, I listed additional questions.

- Q1: In Section 4.4, the authors acknowledged the trade-offs of using l_2 distances. Have the authors considered employing alternative similarity measures, such as cosine similarity? What prompted the decision not to utilize cosine similarity, especially given that the B-cos method inherently leverages this approach?

- Q2: It appears that the negative sign in the L2 distance equation (Eq. 10) may be a typo, as it relates to the process of extracting pseudolabels for each CDF based on the similarity. Is it a typo?

---

> ### Author Response · Authors · 2024-11-20
>
> We thank the reviewer for the comments and constructive feedback. Below we present the clarifications on the weakness and answer the questions.
>
> W1:  The computation of mutual information only involves precomputed features and the respective labels and not the explanation. This is not a computationally intensive process given the precomputed features and can be done offline, before the inference sample is processed. We add more details on the computation of mutual information in the main text.
>
> As shown in figure 2, we divide our method into train and inference. At train time, we train the encoder, compute the train features and then the CDF features. At inference, computation of finding prototype examples takes KNN is a smaller feature space. This space is as small as one feature, which reduces to an argmax over the difference of N pairs of 2 numbers each. We add this to the description.
> We also use the label from the classifier as the pseudo label (added in the appendix). In fact any method that approximates the prediction can serve as a plug in for the getting pseudo label.
> We add another ablation to the appendix to show that the proposed method is robust to the performance of pseudo label prediction mechanism (Figure 12). Please note that using pseudo labels helps create an explanation such that ‘what makes a cat a cat is different from what makes a dog a dog’.
>
> W2: First of all, we would like to spell out the differences between the proposed method and the ProtoPNet family. We would argue (and address it in the text more clearly) that actually ProtoPNet-type models offer post hoc explanation in the following sense. They optimise latent representations of the prototype (in case of ProtoConcept [2], for example, they optimise the parameters of the `prototypical balls’[2].  For these optimised latent representations, they either ‘project/push the training example to the nearest training example‘ (ProtoPNet) or, in ProtoConcept, the authors visualise the training-data prototypes for the concepts contained within a ball.
> In contrast to this, the proposed method visualises the images which are directly forming a part of the decision-making process, thus providing fidelity of the explanations (see Steps 4 and 5 of Algorithm 1). Our method, unlike the suggested baselines, takes decisions based on the explanation (not the other way round). Hence we get a reduced performance but at an advantage of high fidelity <refer table and numbers>. With methods such as B-cosification[5] we can use off the shelf models and then use them for inference in the proposed style, giving high fidelity explanations.
>
> [5] Arya et al (2024) B-cosification: Transforming Deep Neural Networks to be Inherently Interpretable
>
> W3: We use a B-cos network as the backbone and propose an inference method so that the explanation is part of the inference mechanism, i.e. the explanation is not post hoc.
> Instead of explaining the decision that is taken post-hoc through prototypes as it is done in ProtoPNet-type models, the decision making mechanism happens through a ‘this looks like that’-human understandable bottleneck. The trade-off between the inherent prototypical explanations on one hand, and the accuracy on the other hand is therefore the reason behind the inferior performance.
> On the end-to-end model, we see there might be two different interpretations to this: (1) training end-to-end COMiX model or (2) the B-cos backbone without pretraining-and-finetuning, but just trained from random weights. Could the reviewer clarify which one they mean?
> On interpretation (1), we would argue that the there are two class of methods to consider.
> 1) The attribution based explanation like B-cos that gives one global saliency map for a decision. various prior art has shown that this is a suboptimal explanation
> 2) The ‘this looks like that’ family of methods are post hoc in the sense they employ different mechanism for decision and explanation.
> Our method brings the best feature of both these explanation but we think of the the performance of the original B-cos model and the KNN on its feature space as the upper limit of the proposed method. On interpretation (2), we are now checking the performance of training from just random weights.
>
> W4: The original intention was not to show the performance against the competitor, but to show the notional PQ value compared to the standard network. The comparison of sparsity in table is used to further motivate the use of B-cos encoder. Figure 6 shows the much higher sparsity of the proposed method. The performance of the method under as short as 12 features is reported. Also the method is sparse with respect to the number of train samples it uses for decision making (shown with the ablation with different K values). Also in the appendix we show Figure 8 which shows the performance of the method with features considered one at a time.

---

> > ### Author Response · Authors · 2024-11-20
> > **Continued**
> >
> > W5: The type of explanation(this looks like that) presented in our paper has already been validated in the prior art mentioned by you. The novelty of the method is in giving a high fidelity explanation which is completely truthful to the decision making mechanism. That was our original motivation to choose the accuracy and causal metric to evaluate the framework. We understand that the quantitative evaluation in [2] is a user study. We hope not having the user study is not considered a reason for lowering the score.
> >
> > Q1: Thanks for the question. In our revision we also try using cosine distances as they fit well with the Bcos encoder. The results are not significantly different.
> >
> > Q2: No, it’s a maximisation of negative L2 distance instead of minimisation of the L2 distance (arg top_k denotes the generalisation of arg max and not arg min, so we don’t think it’s a typo.

---

> ### Comment · Reviewer_HwsD · 2024-11-27
> **Response by Reviewer HwsD**
>
> - Regarding the response to W1, I have already understood that mutual information computation is performed only in the precomputation phase, which helps to maintain overall computational feasibility. However, the authors' response could have been clearer in addressing my original concern about the mutual information computation process, which remains a key aspect in the proposed method.
>
> - For the responses to W2 and W4, the authors stated that "...the proposed method visualizes the images which are directly forming a part of the decision-making process, thus providing fidelity of the explanations..." However, I find this argument unconvincing. The proposed method is aligned with the fundamental concept of prototypical learning—decisions are made based on the similarity between an arbitrary input and the learned prototypes. Thus, the lack of comparison with other prototype-based methods remains a significant concern. This omission is particularly notable as the authors themselves described the proposed method as a combination of object-level attention and prototypes, as presented in Table 3.
>
> - For the responses to W3 and W5, while the authors emphasized the advantages of using the B-cos network for generating salience maps, prior research [1, 2] has highlighted limitations in the effectiveness of salience maps as meaningful explanations, especially in real-world applications such as medical domains. These limitations should be acknowledged and addressed.
>
> - Thank you for providing clarifications on Q1 and Q2. I hope these points will be more clearly reflected in the revised manuscript.
> Despite the clarifications and responses, I still have several concerns after reviewing the responses and comments from other reviewers. Given the current state of the paper, I will maintain my original rating.
>
> - References
> - [1] Border, Samuel P., and Pinaki Sarder. "From what to why: The growing need for a focus shift toward explainability of AI in digital pathology." Frontiers in Physiology 12 (2022): 821217.
> - [2] Venkatesh, Kesavan, et al. "Gradient-Based Saliency Maps Are Not Trustworthy Visual Explanations of Automated AI Musculoskeletal Diagnoses." Journal of Imaging Informatics in Medicine (2024): 1-10.

---

### Note · Authors · 2025-01-10

I have read and agree with the venue's withdrawal policy on behalf of myself and my co-authors.